# A secondary structure-based position-specific scoring matrix applied to the improvement in protein secondary structure prediction

Teng-Ruei Chen[1,2☯], Sheng-Hung Juan[1,2☯], Yu-Wei Huang[1,2], Yen-Cheng Lin[3,4], Wei-Cheng Lo[1,2,3,4,5]*

1 Institute of Bioinformatics and Systems Biology, National Chiao Tung University, Hsinchu, Taiwan,
2 Institute of Bioinformatics and Systems Biology, National Yang Ming Chiao Tung University, Hsinchu, Taiwan, 3 Department of Biological Science and Technology, National Chiao Tung University, Hsinchu, Taiwan, 4 Department of Biological Science and Technology, National Yang Ming Chiao Tung University, Hsinchu, Taiwan, 5 The Center for Bioinformatics Research, National Yang Ming Chiao Tung University, Hsinchu, Taiwan

☯ These authors contributed equally to this work.
* WadeLo@nctu.edu.tw

**Data Availability Statement:** All relevant data are within the manuscript and its Supporting information files.

## Abstract

Protein secondary structure prediction (SSP) has a variety of applications; however, there has been relatively limited improvement in accuracy for years. With a vision of moving forward all related fields, we aimed to make a fundamental advance in SSP. There have been many admirable efforts made to improve the machine learning algorithm for SSP. This work thus took a step back by manipulating the input features. A secondary structure element-based position-specific scoring matrix (SSE-PSSM) is proposed, based on which a new set of machine learning features can be established. The feasibility of this new PSSM was evaluated by rigid independent tests with training and testing datasets sharing <25% sequence identities. In all experiments, the proposed PSSM outperformed the traditional amino acid PSSM. This new PSSM can be easily combined with the amino acid PSSM, and the improvement in accuracy was remarkable. Preliminary tests made by combining the SSE-PSSM and well-known SSP methods showed 2.0% and 5.2% average improvements in three- and eight-state SSP accuracies, respectively. If this PSSM can be integrated into state-of-the-art SSP methods, the overall accuracy of SSP may break the current restriction and eventually bring benefit to all research and applications where secondary structure prediction plays a vital role during development. To facilitate the application and integration of the SSE-PSSM with modern SSP methods, we have established a web server and stand-alone programs for generating SSE-PSSM available at http://10.life.nctu.edu.tw/SSE-PSSM.

**Funding:** This work was funded by the Ministry of Science and Technology (MOST), Taiwan (https://www.most.gov.tw/?l=en) with grant number NSC 101-2311-B-009-006-MY2 to WCL. The funders had no role in study design, data collection and analysis, decision to publish, or preparation of the manuscript.

**Competing interests:** The authors have declared that no competing interests exist.

## Introduction

The secondary structure prediction of a protein means determining the secondary structural conformation for each residue of the protein merely based on the amino acid sequence. Although SSP has been applied to many fields, its accuracy seems to stay on a plateau of 81–86% for years. We believe that if the accuracy of SSP can be substantially improved, research and applications dependent on it will all be advanced. This work aims to make a fundamental improvement in SSP, hoping that, if the proposed algorithm can be adopted by state-of-the-art SSP methods, the general accuracy of SSP will reach a new level. Thanks to many recent works [1–11], much progress has been brought to the methodology and machine learning algorithms for SSP. Thus, this study focuses on developing a new set of features that can be utilized in all mature machine-learning-based SSP methods. The outcome of our efforts is a new position-specific scoring matrix (PSSM) composed of secondary structural elements instead of amino acid codes.

SSP methods have been developed for more than sixty-five years since Pauling and Corey first proposed the helix and sheet conformations of polypeptides in 1951 [12, 13]. Despite being an "ancient" topic, there have been still more than five methods published every year since 2010 [14] because SSP has many applications in protein sciences, such as fold recognition [15–17], tertiary/quaternary structure prediction and modelling [18–22], functional and evolutionary analyses [23–25], prediction of folding pathway/elements [26, 27], and prediction of disordered regions [28–31], functional sites [32, 33], binding sites [34–36], enzyme target sites [37–39], and suitable bioengineering sites [40–43]. Moreover, predicted secondary structures could be used in *de novo* protein design [44, 45] and protein drug design [46–48].

For carrying out SSP, protein secondary structural conformations were first classified into secondary structure elements (SSEs). There are two widely used sets of SSE: 1) The three-state (Q3) SSEs that describe a protein conformation as helixes (H), strands (E), and coils (C). 2) The eight-state (Q8) SSEs defined by DSSP [49] that include $3_{10}$-helix (G), α-helix (H), π-helix (I), extended β-strand (E), β-bridge (B), turn (T), bend (S), and coil (C). Therefore, the development and evaluation of SSP methods also fall into two major categories, *i.e.*, methods making three-state SSE predictions with Q3 accuracy and methods making eight-state predictions with Q8 accuracy. In general, Q8 methods are also capable of making three-state predictions.

During the past four decades, SSP accuracy has been continuously raised until the last few years. In 1978, Garnier, Osguthorpe, and Robson developed the GOR method based on information theory and achieved ~60% accuracy for helices, β-pleated sheets, reverse turns, and coils [50]. In 1988, Qian and Sejnowski accomplished 64.3% Q3 by neural networks [51]. In 1993, Rost and Sander used multiple sequence alignment (MSA) profiles of homologous sequences as features for a neural network method PHD and increased the Q3 to 69.7% [52]. In 1999, Jones utilized the sequence profile computed by PSI-BLAST, known as the PSSM [53], as features and developed a neural-network-based method PSIPRED, which promoted the Q3 accuracy to 76.5% [1].

After PSIPRED, most SSP methods employed PSSM as the major feature set, and the competition of accuracy had then mainly focused on the algorithm and improvement in machine learning, such as neural networks [54–57], support vector machine [58–60], and hidden Markov models [56, 61]. The Q3 accuracy of methods developed in the early 2000s generally fell between 76% and 78% [62]. Meanwhile, the Q8 accuracy of SSpro8 broke the record and reached 62.6% [55]. In 2007, the Q3 came up to 79.5% when Dor and Zhou developed the SPINE based on neural networks and a large training dataset [63]. Soon after that, Jpred 3 [64], another neural-network-based method, broke 80% Q3 in 2008. Since then, neural network derivatives have formed the core of most SSP methods. Because of the rapid growth of

computing power and the amount of available training data (proteins with determined structures), the architectures of neural networks became increasingly sophisticated, inclusive of the BRNN [65], CNF [7], BRNN-LSTM [66], deep learning [8–10] and CRNN [11]. However, the accuracy seems to meet a plateau; little improvement has been made during the past ten years. The Q3 of current SSP methods evaluated with strict independent tests falls between 81% and 86%, while there has not yet been Q8 higher than 75%.

Since the PSSM was introduced into this field, few changes have been made to the fundamental feature set. Theoretically, the quality of features (and training data) should be the ultimate limitation for the accuracy of a predictor, suggesting that if the feature set used by current methods could be improved, the overall accuracy of SSP would be enhanced. We thus proposed an SSE-based PSSM that might be easily integrated into various SSP methods to achieve such enhancements. The way we computed this new PSSM was just a few steps away from the traditional PSSM, the same as which the values of the proposed PSSM could be readily used as features for machine learning. In order to make a reliable evaluation of the SSE-PSSM, a careful experimental dataset preparation procedure was conducted such that any two datasets for either training, testing, or independent tests shared <25% sequence identities. Independent tests revealed that, by using the same machine learning strategy, the accuracy accomplished by the SSE-PSSM feature set was much higher than that by the conventional amino acid PSSM feature set. Preliminary tests, in which the SSE-PSSM feature set was indirectly incorporated into state-of-the-art SSP methods, showed that these methods' accuracy could be significantly improved. For instance, when it was preliminarily integrated with DeepCNF [7], Q3 raised from 82.9% to 84.1%, and Q8 from 68.4% to 71.8%, as assessed with the CASP12 independent set [14, 67]. We aimed to bring a general improvement to SSP by updating the machine learning feature set, and the proposed SSE-PSSM has been demonstrated promising for this purpose.

## Results

### Example of the SSE-PSSM

The main contribution of this work is an SSE-based PSSM that may serve SSP algorithms as a new feature set. Fig 1 demonstrates the difference between a traditional amino acid-based PSSM and the proposed SSE-PSSM. The first advantage of the SSE-PSSM is its simplicity of implementation. With a protein sequence similarity search program like the PSI-BLAST [53] or HHBlits [68] and a protein structure dataset, a set of aligned amino acid sequences can be transformed into aligned SSE sequences and construct the SSE-PSSM. Another advantage lies in the density of information. The intermediate of a PSSM is the position propensity matrix (PPM). If the query protein is so novel that few homologs exist in the target dataset, the PPM generated from the aligned amino acid sequences may contain too many zeros to maintain the quality of the generated PSSM. Because the number of SSE codes (3 or 8) was much smaller than that of amino acids (20), with the same amount of homologs, the SSE-PPM will have much fewer zeros and hence a higher quality of the PSSM. Besides, the small number of SSE codes is feasible to improve the efficiency of machine learning algorithms, the time cost of which usually increases as the number of input features increases [69].

Typically, a sequence alignment block is transformed into a position propensity matrix and then the PSSM. In generating the proposed SSE-based PSSM, the main difference is an additional SSE transformation step (see Materials and methods). By introducing secondary structure information from the query protein's homologs into the system, the SSE-PSSM feature set helps improve SSP accuracy. The SSE transformation decreases the number of codes and significantly reduces the proportion of null entries in the propensity matrix, which eventually

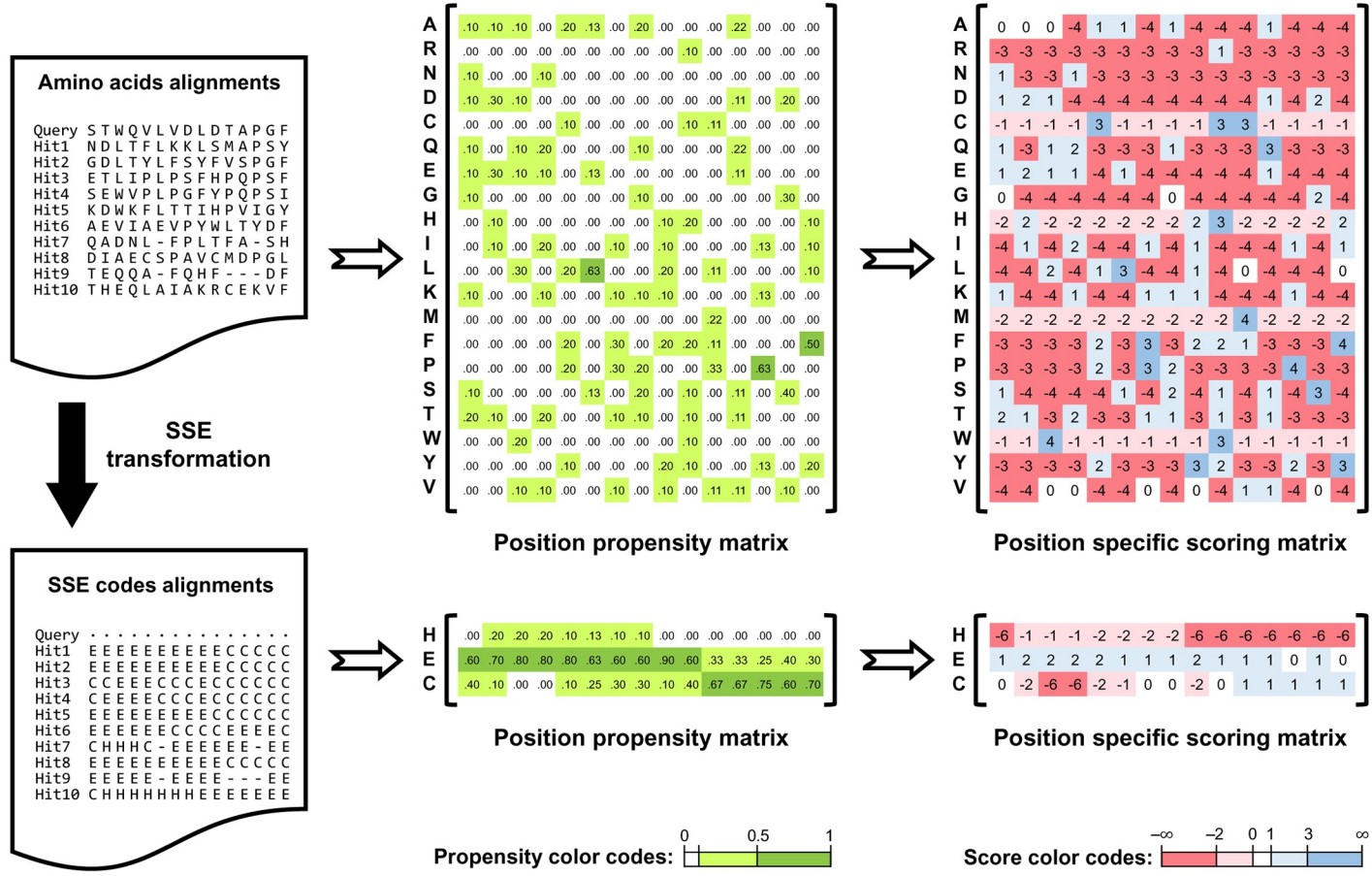

**Fig 1. Position-specific scoring matrix of amino acids *vs*. secondary structure codes.**

enhances the quality of the final scoring matrix and may further improve the prediction accuracy. This illustration was made using part of the alignment of a real case, a Nudix hydrolase (PDB 1mqw). For simplicity, three-state SSE codes were applied here. In this study, the actual predictive models of the SSE-PSSM feature set were constructed using eight-state codes.

## Accuracy of secondary structure prediction by SSE-PSSM

The SSE-PSSM is a **feature set** rather than a predictor. In order to evaluate it, an integrated machine learning system we developed [41, 42] was utilized. Since the focus of this study is the new feature set instead of machine learning, we simply adopted the default settings of this system without optimization. As a basis for comparison, the traditional amino acid PSSM (AA-PSSM) feature set was also generated. The predictive models of SSE- and AA-PSSMs were trained with QuerySet-T, and independent tests were performed with QuerySet-I. For avoiding the bias of prediction and information leakage, these datasets were iteratively homology reduced such that any two proteins from them shared <25% identity (see **"Experimental datasets"**).

As shown in Table 1, no matter in the three- or eight-state prediction, the overall accuracy (Q3 and Q8), boundary accuracy, and internal accuracy of SSE-PSSM were all higher than those of AA-PSSM; and the misclassification rates of SSE-PSSM were considerably lower. The SOV (segment overlap measure), a more critical assessment measure than the Q accuracy, can

**Table 1. Accuracy of the SSE-PSSM and several state-of-the-art SSP methods.**

| SSE set | Method | Performance (QuerySet-I against TargetSet-nr25) | | | | | | |
|---|---|---|---|---|---|---|---|---|
| Three-state | **Measure** | **Q3** | **SOV3** | **Boundary Q3** | **Internal Q3** | **H↔E err** | **H↔C err** | **E↔C err** |
| | SSE-PSSM | 0.785 | 0.742 | 0.692 | 0.830 | 0.034 | 0.107 | 0.074 |
| | AA-PSSM | 0.654 | 0.553 | 0.546 | 0.706 | 0.099 | 0.139 | 0.108 |
| | Scorpion | 0.748 | 0.716 | 0.612 | 0.814 | 0.037 | 0.114 | 0.101 |
| | Spider2 | 0.755 | 0.688 | 0.625 | 0.818 | 0.033 | 0.112 | 0.100 |
| | SpineX | 0.742 | 0.692 | 0.597 | 0.812 | 0.034 | 0.121 | 0.104 |
| | PSIPRED | 0.751 | 0.694 | 0.612 | 0.818 | 0.023 | 0.124 | 0.102 |
| | DeepCNF | 0.761 | 0.716 | 0.630 | 0.826 | 0.031 | 0.105 | 0.103 |
| | RaptorX | 0.733 | 0.675 | 0.596 | 0.800 | 0.046 | 0.114 | 0.108 |
| | SSpro8 | 0.725 | 0.661 | 0.581 | 0.795 | 0.048 | 0.121 | 0.107 |
| Eight-state | **Measure** | **Q8** | **SOV8** | **Boundary Q8** | **Internal Q8** | **Hs↔Es err[a]** | **Hs↔Cs err[a]** | **Es↔Cs err[a]** |
| | SSE-PSSM | 0.663 | 0.657 | 0.555 | 0.773 | 0.036 | 0.114 | 0.074 |
| | AA-PSSM | 0.479 | 0.446 | 0.341 | 0.564 | 0.120 | 0.154 | 0.114 |
| | DeepCNF | 0.642 | 0.587 | 0.487 | 0.801 | 0.031 | 0.105 | 0.103 |
| | RaptorX | 0.617 | 0.553 | 0.451 | 0.786 | 0.046 | 0.114 | 0.108 |
| | SSpro8 | 0.605 | 0.548 | 0.432 | 0.781 | 0.048 | 0.121 | 0.107 |

[a]These misclassification rates were computed by reducing the eight SSE codes of DSSP to three codes, helices (H, G, and I), strands (E and B), and coils (C, S, and T).

effectively capture the overall quality of SSP for a protein and reduce noises from individual residues [70–72]. Previous studies calculated the SOV on the three-state basis (SOV3). Here a more critical eight-state SOV, the SOV8, was also calculated. Judging from the SOV data, the same conclusion was reached that SSE-PSSM outperformed the traditional AA-PSSM as an SSP feature set.

Several state-of-the-art SSP methods were tested. Although the SSE-PSSM model achieved higher scores in most measures, it is noteworthy that the SSE-PSSM and those methods were not applied on an equal basis. In this experiment, except for the SSE-PSSM model, all methods were developed based on the traditional PSSM. This experiment aimed to demonstrate the feasibility of the SSE-PSSM as an **SSP feature set** but not to compare the accuracy among SSP methods. The machine learning system we utilized to establish the SSE- and AA-PSSM models was only a simple setup. If the advanced machine learning algorithms of those SSP methods could be applied, the accuracy of the AA-PSSM model should be at a level similar to them. Taking the accuracy of the AA-PSSM as a baseline, the SSE-PSSM achieved >12% and >18% improvements in Q3 and Q8, respectively. If the SSE-PSSM could be adopted by those state-of-the-art methods, their accuracy would be significantly improved as well, pushing much forward the leading edge of SSP. The Q8 of several recent works may break the current upper level of 75% [5–7].

The accuracies reported here were lower than those in previous reports because the way we prepared datasets was so stringent that it limited the size of the PSSM target dataset TargetSet-nr25, which consisted of only 11.4 thousand sequences. The size of target dataset has been shown to exhibit a positive correlation with SSP accuracy [73]. When a conventional Uni-Ref90-2015 of 38.2 million sequences was applied, the accuracies were all at the same levels as previously reported (*i.e.*, ~81% Q3; see "**Applied secondary structure prediction methods**"). To ensure the reliability of the results based on the small TargetSet-nr25, we repeated all

**Table 2. Accuracy of several SSP methods for proteins of different structural classes.**

| SSE set | Method | All-alpha proteins (SCOP class a) | | All-beta proteins (SCOP class b) | | Alpha-beta mixed proteins (SCOP class a/b) | | Alpha-beta segregated proteins (SCOP class a+b) | | Normalized standard deviation[a] | |
|---|---|---|---|---|---|---|---|---|---|---|---|
| Three-state | Measure | Q3 | SOV3 | Q3 | SOV3 | Q3 | SOV3 | Q3 | SOV3 | Q3 | SOV3 |
| | SSE-PSSM | 0.775 | 0.699 | 0.764 | 0.733 | 0.819 | 0.794 | 0.781 | 0.741 | 0.029 | 0.050 |
| | AA-PSSM | 0.638 | 0.463 | 0.641 | 0.584 | 0.681 | 0.619 | 0.655 | 0.554 | 0.029 | 0.108 |
| | Scorpion | 0.765 | 0.718 | 0.709 | 0.671 | 0.771 | 0.758 | 0.744 | 0.714 | 0.036 | 0.047 |
| | Spider2 | 0.772 | 0.663 | 0.705 | 0.642 | 0.775 | 0.732 | 0.749 | 0.690 | 0.042 | 0.053 |
| | SpineX | 0.761 | 0.699 | 0.692 | 0.646 | 0.760 | 0.732 | 0.735 | 0.693 | 0.042 | 0.048 |
| | PSIPRED | 0.774 | 0.706 | 0.712 | 0.660 | 0.764 | 0.730 | 0.740 | 0.696 | 0.036 | 0.040 |
| | DeepCNF | 0.791 | 0.725 | 0.726 | 0.682 | 0.780 | 0.759 | 0.756 | 0.724 | 0.036 | 0.042 |
| | RaptorX | 0.757 | 0.673 | 0.694 | 0.657 | 0.757 | 0.725 | 0.735 | 0.690 | 0.039 | 0.040 |
| | SSpro8 | 0.753 | 0.667 | 0.681 | 0.641 | 0.745 | 0.711 | 0.725 | 0.678 | 0.043 | 0.041 |
| Eight-state | Measure | Q8 | SOV8 | Q8 | SOV8 | Q8 | SOV8 | Q8 | SOV8 | Q8 | SOV8 |
| | SSE-PSSM | 0.657 | 0.636 | 0.624 | 0.626 | 0.711 | 0.710 | 0.659 | 0.658 | 0.051 | 0.053 |
| | AA-PSSM | 0.478 | 0.374 | 0.417 | 0.426 | 0.496 | 0.487 | 0.456 | 0.430 | 0.069 | 0.095 |
| | DeepCNF | 0.649 | 0.617 | 0.545 | 0.526 | 0.619 | 0.621 | 0.587 | 0.583 | 0.069 | 0.071 |
| | RaptorX | 0.617 | 0.572 | 0.498 | 0.491 | 0.597 | 0.586 | 0.566 | 0.557 | 0.084 | 0.072 |
| | SSpro8 | 0.62 | 0.580 | 0.479 | 0.477 | 0.593 | 0.588 | 0.558 | 0.553 | 0.099 | 0.086 |

[a]The standard deviation of an accuracy measure was normalized by dividing it by the maximum of the measure values obtained with the same predictor. The purpose of this normalization is to present clearly for each predictor the relative range of fluctuation of accuracies.

experiments three times by random sampling to obtain the averaged performance (see "**Experimental datasets**").

## Performance for various structure classes

To our knowledge, the accuracy of SSP methods on different protein structural classes has not been discussed. Here we grouped the query proteins of our independent test set (QuerySet-I) according to their SCOP (Structural Classification of Proteins) classes [74] to analyze the performance of the predictive model of SSE-PSSM and several SSP methods. Table 2 demonstrated that most SSP methods were most sensitive to either alpha-beta mixed or all-alpha proteins if the accuracy was determined based on residues (Q3/Q8). Interestingly, all predictors showed the best performance for alpha-beta mixed proteins when measuring the accuracy according to secondary structure segments (SOV3/SOV8; see Discussion). In implementing the SSE-PSSM as an SSP feature set, attention might be paid to all-alpha proteins, for it performed relatively weak in this class. The DeepCNF, an excellent algorithm based on deep convolutional neural fields [7], outperformed the SSE-PSSM model even with a traditional PSSM as its main feature set. Nevertheless, an advantage the SSE-PSSM may bring to current SSP methods is the balanced performance, for it achieved the smallest standard deviations of Q3, Q8, and SOV8 among these structural classes.

An underline indicates that the SSP method performed best in this class compared to its accuracy in these four classes.

These results also revealed an issue: all methods performed the poorest when predicting all-beta proteins. Although it had been reported that most SSP methods performed worse in strands than in helices [4, 14], this is perhaps the first time SSP methods were challenged with proteins overall folded into strands *versus* helices, and the poor performance in all-beta proteins should be informative for future SSP developments. One way out of this problem might

**Table 3. Accuracy of several SSP methods for proteins of different sizes.**

| SSE set | Method | Small proteins (<150 residues) | | Medium-sized (150–299 residues) | | Large proteins (≥300 residues) | | Normalized standard deviation | |
|---|---|---|---|---|---|---|---|---|---|
| Three-state | Measure | Q3 | SOV3 | Q3 | SOV3 | Q3 | SOV3 | Q3 | SOV3 |
| | SSE-PSSM | 0.791 | 0.765 | 0.810 | 0.769 | 0.769 | 0.720 | 0.025 | 0.035 |
| | AA-PSSM | 0.678 | 0.592 | 0.667 | 0.570 | 0.640 | 0.534 | 0.029 | 0.049 |
| | Scorpion | 0.777 | 0.779 | 0.764 | 0.736 | 0.731 | 0.687 | 0.031 | 0.059 |
| | Spider2 | 0.774 | 0.738 | 0.763 | 0.693 | 0.738 | 0.660 | 0.024 | 0.053 |
| | SpineX | 0.765 | 0.752 | 0.752 | 0.710 | 0.723 | 0.667 | 0.028 | 0.057 |
| | PSIPRED | 0.781 | 0.749 | 0.764 | 0.728 | 0.731 | 0.668 | 0.033 | 0.056 |
| | DeepCNF | 0.791 | 0.776 | 0.783 | 0.754 | 0.746 | 0.691 | 0.030 | 0.057 |
| | RaptorX | 0.776 | 0.748 | 0.752 | 0.711 | 0.718 | 0.656 | 0.038 | 0.062 |
| | SSpro8 | 0.759 | 0.73 | 0.741 | 0.694 | 0.709 | 0.649 | 0.033 | 0.056 |
| Eight-state | Measure | Q8 | SOV8 | Q8 | SOV8 | Q8 | SOV8 | Q8 | SOV8 |
| | SSE-PSSM | 0.678 | 0.674 | 0.692 | 0.678 | 0.643 | 0.641 | 0.036 | 0.030 |
| | AA-PSSM | 0.509 | 0.478 | 0.475 | 0.434 | 0.443 | 0.412 | 0.065 | 0.070 |
| | DeepCNF | 0.650 | 0.643 | 0.627 | 0.613 | 0.574 | 0.558 | 0.060 | 0.067 |
| | RaptorX | 0.624 | 0.614 | 0.594 | 0.571 | 0.544 | 0.525 | 0.065 | 0.072 |
| | SSpro8 | 0.615 | 0.606 | 0.587 | 0.571 | 0.537 | 0.523 | 0.064 | 0.069 |

be a two-staged approach in which the structural class of the query sequence is predicted at first, and then the per-residue secondary structure is predicted by using predictive models specifically established for the predicted structural class.

## Performance for different protein sizes

It is not clear whether current SSP methods perform equally for proteins of different sizes. The independent test dataset (QuerySet-I) was divided into three size subsets. As revealed by Table 3, except for the SSE-PSSM predictive model, the performance of all predictors dropped as the size increased. The SSE-PSSM performed best for medium-sized proteins, whereas all other methods performed best for small ones (see the underlined). Judging from the standard deviation of accuracy, the SSE-PSSM had a relatively stable performance regarding the size of proteins. This balance was not just the effect of how we trained the predictive model of SSE-PSSM, because the model of AA-PSSM (trained with the same data and machine learning procedure) performed best for small and worst for large proteins as the amino acid PSSM methods did. Thus, adopting the SSE-PSSM feature set may help current SSP methods balance the performance for proteins of different sizes.

## Improvement in accuracy by combining SSE-PSSM with traditional PSSM and state-of-the-art SSP methods

This study constructed two types of feature sets, the proposed SSE-PSSM and the widely used AA-PSSM. For an SSP method using AA-PSSM already, we wondered whether its accuracy would be enhanced if the SSE-PSSM were incorporated into the feature set. As a pretest, using the same machine learning system [41, 42], we constructed a predictive model by combining the AA- and SSE-PSSMs. This model was trained with QuerySet-T. In addition to our independent test dataset (QuerySet-I), three independent datasets established in previous studies, the TS115, CASP12, and CASP13 [14, 67], and their companion target dataset UniRef90-2015 were also recruited to make rigorous assessments. As shown in S1 Table, after the feature set

integration, the accuracy on all datasets was remarkably improved over solely using the AA-PSSM.

The success in improving the accuracy of the traditional AA-PSSM feature set by combining it with the SSE-PSSM encouraged us to take a step further to incorporate the SSE-PSSM into state-of-the-art SSP methods. However, the unavailability of the source codes of the programs made it impossible to change the feature set of those methods. Thus, we used a trick to construct new predictive models by using the output of the programs as input features. For verifying the feasibility of this "second-level" machine learning strategy, both the results of training and independent test of constructed models were compared with the raw results of these methods. As demonstrated in Materials and methods, this second-level learning did not much influence their accuracy. With this strategy, additional features like the SSE-PSSM could be "indirectly" integrated into a well-developed predictor. As diagramed in Fig 2, independent tests concluded that the predictive models of these state-of-the-art SSP methods integrated with the SSE-PSSM feature set all outperformed the models without integration. After the integration, in general, eight-state predictions were improved more significantly than three-state ones. On average, the Q3/SOV3 and Q8/SOV8 were improved by 3–4% and ~7%, respectively (see also S2 Table for detailed data). Taking only the results of third-party independent datasets (TS115 and the CASPs) into consideration, the improvements in Q3/SOV3 and Q8/SOV8 still reached 2.0%/2.9% and 5.2%/5.1%, respectively. As a control test, the traditional AA-PSSM had also been integrated in the same way into those methods. Because the original predictive models of those methods were all constructed with the traditional PSSM, this feature set integration was expected to bring little benefit. Supported by the results listed in S3 Table, most SSP accuracy values of these methods increased less than 1% while some even decreased.

Comparing the significant differences between integrating SSE-PSSM and the traditional AA-PSSM, SSE-PSSM was demonstrated feasible to improve the accuracy of current SSP methods. For precisely determining the extent to which SSE-PSSM can enhance accuracy, this indirect feature integration experiment was just a preliminary test. Future collaboration for implementing the SSE-PSSM feature set into state-of-the-art SSP methods is in demand to directly examine the feasibility of SSE-PSSM.

## Improvement in accuracy by combining SSE-PSSM with state-of-the-art SSP methods that depend on HHBlits PSSM

After this study was initiated, more and more SSP algorithms utilized the HHBlits [68] to perform MSA analysis between the query protein and target dataset proteins, encode the resultant conservation profile as a PSSM, and make SSP either directly by the HHBlits-PSSM (*e.g.*, NetSurfP-2 [10]) or by both the PSSMs generated by HHBlits and PSI-BLAST (*e.g.*, MUFOLD-SS [9], Porter5 [11], and Spider3 [8]). To examine whether SSE-PSSM is compatible with the HHBlits-based SSP methodology, we first implemented it based on HHBlits MSA and assessed it by 1) using it along as the predictive feature set, 2) combining it with the AA-PSSMs produced with HHBlits and PSI-BLAST (AA-PSSMhh for short) as a large feature set, and 3) preliminary integrating it with several HHBlits-based SSP algorithms in the same way stated above. As summarized in S4 Table, the average Q3/Q8 on TS115 and the CASP datasets of the HHBlits-based SSE-PSSM (SSE-PSSMhh) and AA-PSSMhh was 77.9%/63.3% and 73.8%/52.8%, respectively. After combining with the SSE-PSSMhh, the Q3/Q8 and SOV3/SOV8 of AA-PSSMhh was respectively improved by 7.8%/12.5% and 9.0%/12.5% on average. As demonstrated in Fig 3, the four HHBlits-based SSP algorithms mentioned above were preliminarily integrated with the SSE-PSSMhh feature set (refer to S4 Table for raw data). Before the feature integration, the average Q3 and Q8 of those algorithms on TS115 and the CASPs were 84.1%

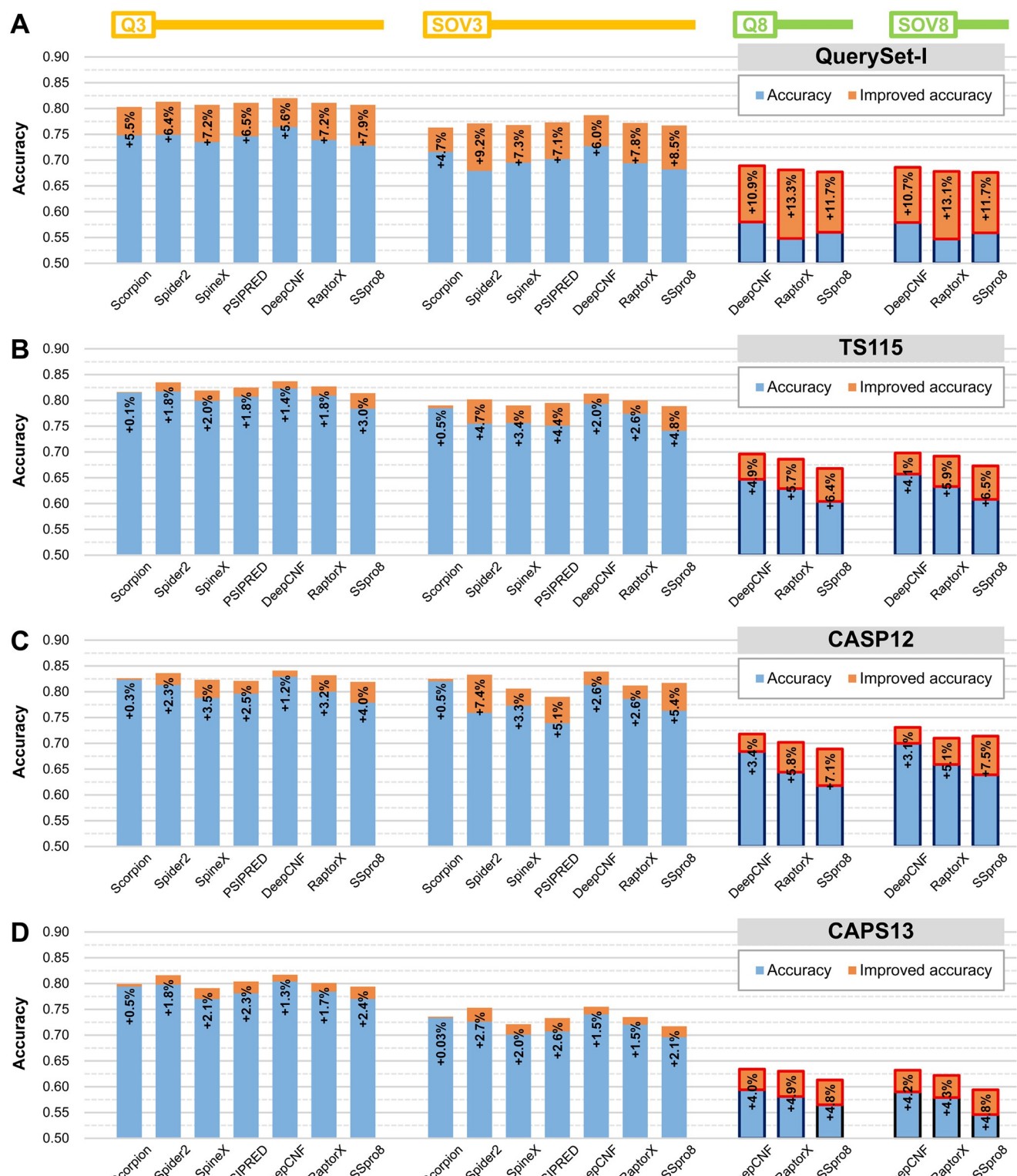

**Fig 2. Performance of preliminary incorporation of SSE-PSSM into state-of-the-art SSP methods using PSI-BLAST to generate PSSM.** (A) QuerySet-I against TargetSet-nr25, the developmental PSSM target dataset of this study. (B) TS115 against UniRef90-2015, the standard PSSM target dataset used in most SSP works. (C) CASP12 against UniRef90-2015. (D) CASP13 against UniRef90-2015. The SSE-PSSM was preliminarily incorporated with different SSP methods using a second-level machine learning feature set integration strategy. After the feature integration, the accuracy of most methods was significantly improved, especially in Q8 and SOV8. The fundamental prediction feature set of all the tested SSP methods are the amino acid PSSM generated by PSI-BLAST.

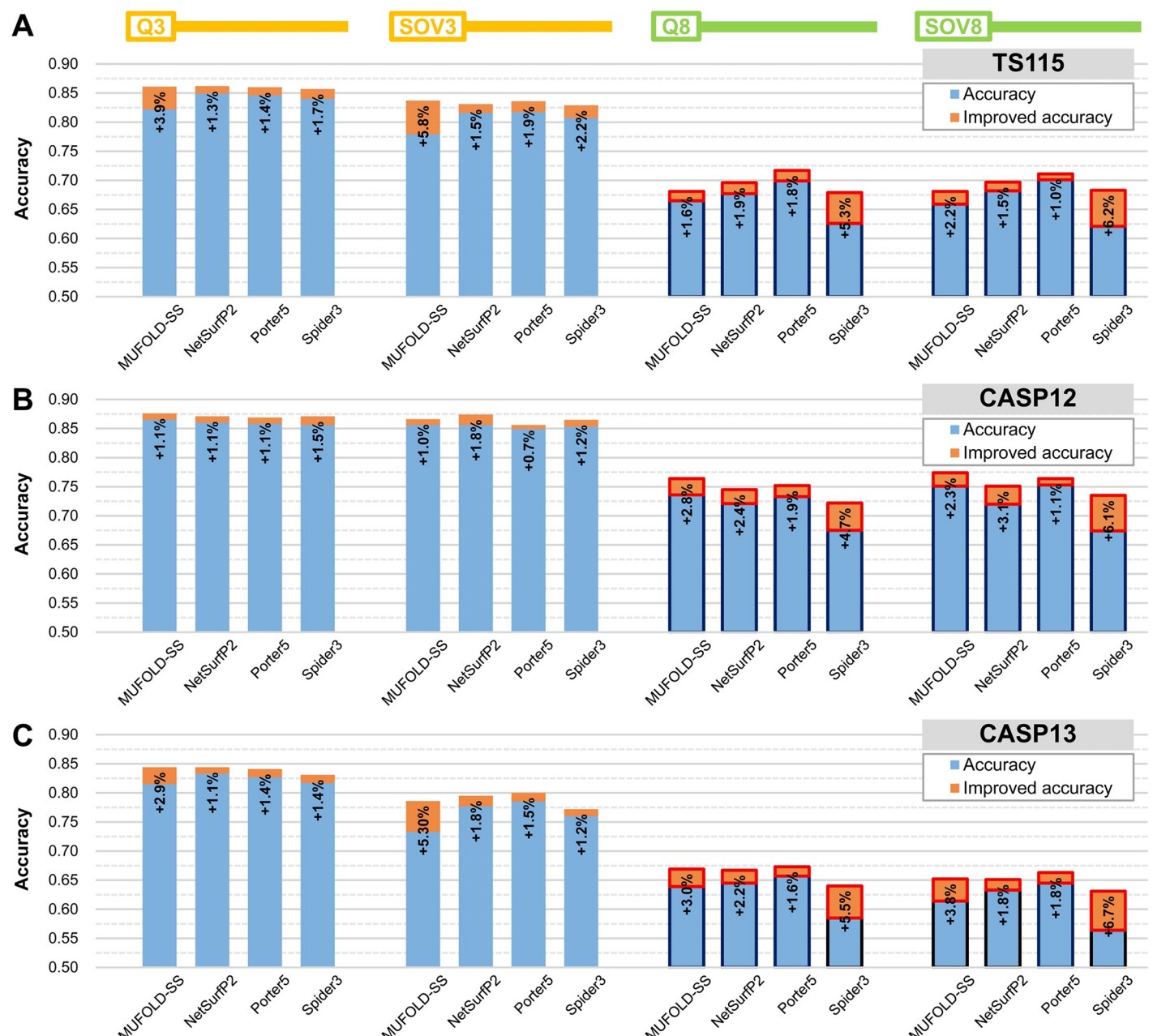

**Fig 3. Performance of preliminary incorporation of SSE-PSSM into state-of-the-art SSP methods using HHBlits to generate PSSM.** (A) TS115 against the UniRef90-2015 PSSM target dataset. (C) CASP12 against UniRef90-2015. (D) CASP13 against UniRef90-2015. These methods used HHBlits as the main PSSM generator. Except for NetSurfP-2, the PSI-BLAST PSSM was also applied in their algorithms. We used both HHBlits and PSI-BLAST to implement the SSE-PSSM and preliminarily integrate it with these HHBlits-based SSP methods. Their accuracies were higher than those of the algorithms tested in Fig 2. However, since their programs were released after 2017 and TS115 and CASP12 proteins were released before 2017, they might have learned some homologs of these datasets. Thus, the CASP13, which comprised proteins released between 2017 and 2019, should be the most reliable independent test dataset among the three. Assessed with CASP13, the preliminary feature integration of SSE-PSSM into these HHBlits-based methods improved the Q3 and Q8 by 1–3% and 2–6%, respectively.

and 67.2%, respectively. After integration, the average Q3 and Q8 increased to 85.7% (+1.6%) and 70.0% (+4.3%), respectively. In general, the performance of these HHBlits-based algorithms in SOVs was weaker than in Q measures. The improving effect of SSE-PSSMhh for the assessed four SSP algorithms on the SOV3/8 accuracies was +2.2%/+3.1% on average.

## Discussion

### Category of a predictor established with the SSE-PSSM

According to whether a template structure is utilized to infer the per-residue conformation of a query protein, SSP algorithms can be classified into two categories: template-based and template-free [6]. Since the critical step of the proposed version of SSE-PSSM algorithm is to determine the SSE sequences of the query protein's homologs based on a reference structure dataset, an SSP method developed using this SSE-PSSM is essentially a template-based method. Nevertheless, there are some differences between the SSE-PSSM feature set and classic template-based methods. Firstly, an extensively aligned homolog of the query protein is not required. If the query protein is unique or if the target dataset consisted of highly non-redundant sequences, the homologs retrieved by the sequence search engine can be very divergent, and the aligned regions between the query protein and homologs are highly fragmented. Even in such cases, the SSE-PSSM can still be assembled. Secondly, the SSE-PSSM scores of a residue position are determined by the substitution rates of conformations, not by the conformations of residues aligned with it.

In this work, the query and target datasets for training and independent tests were all highly non-redundant. The non-redundancy not only existed between datasets but also within every dataset. For instance, QuerySet-I and TargetSet-nr25 are 25% identity non-redundant in between; meanwhile, in QuerySet-I, any two sequences shared <25% identity, so did sequences in TargetSet-nr25. Therefore, it should be tough for any query in our experiments to find closely related homologs to be its template. It is more likely that the SSE-PSSM of a query was constructed entirely based on marginally aligned far-related homologs. The high accuracy accomplished by the predictive model of SSE-PSSM indicated that SSE-PSSM did not rely on high-quality templates.

For comparison, we performed accuracy assessments and feature integration tests on the template-based version SSpro8 (SSpro8T) using the same independent datasets as well as the testing dataset of SSpro8T [6] (abbreviated as TSsspro8 in this paper). SSpro8T exhibited different properties from the eleven template-free algorithms we utilized. First, its accuracy dropped rapidly as the independent dataset became more updated, supposedly because of the increasing difficulty in finding good templates for new novel proteins that are very different from old PDB (Protein Data Bank) [75] releases. For instance, as listed in S5 Table, the macro-average Q3 of the seven PSI-BLAST-based algorithms on TSsspro8 and CASP12 was 79.8% and 79.5%, and that of the four HHBlits-based algorithms was 85.5% and 84.7%, respectively. Both types of template-free algorithms exhibited a <1% drop in accuracy. However, the macro-average Q3 of SSPro8T on TSsspro8 and CASP12 was 94.4% and 78.9%, a 15.5% drop. As for the predictive model of SSE-PSSM, its Q3 on these datasets was 86.2% and 79.0% (S1 Table), equivalent to a 7.2% drop. This fact supported that an SSP method working based on the SSE-PSSM is essentially template-based. Second, when SSpro8T was integrated with SSE-PSSM, the accuracy improvement in Q3/SOV3 was negligible (<0.1%, see S2 Table), no matter assessed with any dataset. The Q8/SOV8 on most datasets were even reduced. It has been demonstrated that all the eleven template-free algorithms would significantly gain accuracy once integrated with SSE-PSSM. To determine whether this difference between SSpro8T and template-free methods resulted from the opposite properties of template-based and template-free algorithms or it was because of the weakness of SSE-PSSM, we applied the same feature-integration procedure to combine SSpro8T with DeepCNF, the most accurate PSI-BLAST-based template-free method we utilized, and PSIPRED, the rapidest one. The results summarized in S2 Table, where the Q3/SOV3 of SSpro8T integrated with either DeepCNF or PSIPRED were barely improved and the Q8/SOV8 decreased, revealed that the SSE-PSSM is

analogous to template-free SSP methods in some aspects. We speculated that, when the integrated feature set of SSpro8T and SSE-PSSM (or a template-free method) was trained with the QuerySet-T, which consisted of PDB structures deposited before 2015, the SSpro8T would dominate the accuracy because it was trained by structures deposited before or in 2013, rather close to the deposit year of QuerySet-T proteins. Once the predictive model was established, features other than SSpro8T would be ineffective or even become noise in actual predictions.

To sum up, an SSP method developed based on the template-based SSE-PSSM can significantly outperform template-free methods when good template(s) are available; otherwise, it will still work by assembling fragmented far-related templates and produce analogous results of template-free methods.

## From where SSE-PSSM brings improvements

An apparent reason why the SSE-PSSM could help current SSP methods achieve improved accuracies is the usage of secondary structure information of the homologs of the query protein. According to the alignment of homologs, as a traditional amino acid PSSM provides per residue the conservation pattern of amino acids [53], an SSE-PSSM provides the conservation pattern of secondary structure elements, which is supposed much more referable in SSP than the former. Another reason for the improvement might be the high information density of the matrix, as demonstrated by the fact that the PPM of an SSE-PSSM may contain fewer null data than an AA-PSSM (Fig 1). To examine the influence of the information density, we computed the average proportion of codes with zero occurrence frequency in the PPM of the AA-PSSM and SSE-PSSM. As shown in S6 Table, the proportion of null-occurrence codes of the SSE-PSSM was remarkably lower than that of the AA-PSSM. If the proportion of null-occurrence did influence the performance of a PSSM in SSP, when the size of the SSE alphabet increased, the accuracy should decrease. As expected, when the SSE alphabet was changed from the DSSP alphabet (8 codes) to the kappa-alpha (22 codes) [76] or the SARST alphabet (23 codes) [77], as the proportion of null-occurrence codes increased, the SSP accuracy dropped.

## On the accuracy of secondary structure prediction for proteins of different structural classes and sizes

Among helix, strand, and coil conformations, SSP methods' accuracy is usually highest in helices and lowest in strands [4, 14]. Similar results were observed in Table 2. The accuracy of all applied state-of-the-art SSP methods yielded the highest or second-highest Q3/8 in all-alpha (mainly helices) and the lowest Q3/8 in all-beta (mainly strands) proteins. Interestingly, when measuring the accuracy using SOV3/8 instead of Q3/8, the most inferior accuracy remained in the all-beta class; however, the highest changed to the alpha-beta mixed class. We speculate that the reason is the property of the SOV algorithm, which is more sensitive to whether the pattern of SSE segments is correctly predicted than whether each SSE code is precisely positioned [71]. Because of it, if the boundaries of SSE segments were predicted imperfectly, the decrease of SOV in all-alpha proteins may be greater than that in alpha-beta mixed. The traditional Q3/8, on the contrary, is sensitive to the boundary of SSE segments because it counts every residue position equally. As a result, imprecise boundary predictions will decrease the Q3/8 more in alpha-beta mixed than in all-alpha proteins, as exemplified below,

```
All-alpha,              Q3 = 0.600, SOV3 = 0.441
Actual     CHHHHCHHHHCHHHHCHHHHCHHHH
Predicted  HHCHHHHCHHHHCHHHHCHHHHCHH
Alpha-beta mixed,       Q3 = 0.440, SOV3 = 0.526
Actual     CHHHHCEEEECHHHHCEEEECHHHH
```

```
Predicted  HHCHHHHCEEEECHHHHCEEEECHH
Alpha-beta segregated    Q3 = 0.520, SOV3 = 0.462
Actual     CHHHHCHHHHCHHHHCEEEECEEEE
Predicted  EECHHHHCHHHHCHHHHCEEEECEE
```

where the syntaxes of these pseudo secondary structures are the same (−**C**XXXX−), and the only difference between an actual and a predicted SSE sequence is a frameshift. With the same secondary structural segment patterns, the fluctuations of Q3 are higher than those of SOV3. When Q3 of the alpha-beta mixed case is so much lower than Q3 of the all-alpha case (0.440 < 0.600), SOV3 of the alpha-beta mixed case is even higher than SOV3 of the all-alpha case (0.526 > 0.441). In addition to the different algorithmic properties regarding imprecise boundaries, these examples reveal that Q and SOV have very different tendencies for proteins of different structural types. We would like to suggest a cautious use of both measures in situations where the accuracy of an SSP method is assessed with multiple datasets, especially when the contents of protein structure classes of those datasets vary.

The formation of a β-strand-pleated sheet may involve residues distant in a protein sequence, while residues of a helix are usually located nearby. Therefore, most previous studies explained that the higher prediction accuracy in helices than strands is because of the difficulty of catching the long-range residue interactions of strands [14, 62]. Here we speculate that imbalanced training cases may be another reason. The natural prevalence of strands is lower than that of helices (around 3:5). In the development of an SSP predictor, if the balance of the structural classes of cases were not considered, it is very likely that the training datasets contain fewer strands than helices and hence make the predictor less sensitive to strands. The first supporting data are the relatively balanced performances of the SSE-PSSM model in different structure classes (Table 2), which might be accomplished partly because of the SCOP class-balanced training dataset (QuerySet-T; Materials and methods). Indirect support comes from the balanced performance of the SSE-PSSM model for proteins of different sizes (Table 3), which might be the effect of the fact that QuerySet-T was also a size-balanced dataset. Hence, we suggest future SSP studies use the approach we prepared the training dataset to balance the SSP performance for various types of proteins.

## Limitation of the proposed approaches

An understandable disadvantage of the current SSE-PSSM algorithm is that the SSE sequences of target proteins are required. When the structure of a hit retrieved by sequence similarity search from the target dataset is not available, our solution was to run a second-round similarity search against a reference dataset of non-redundant structures to construct an approximate SSE sequence for the hit (Subsection: **Algorithm of the SSE-PSSM**). Although this approximation indeed helped improve the accuracy of all applied SSP methods as the SSE-PSSM was incorporated, the improvements were smaller than situations where the target protein structures were available (compare the TargetSet-nr25 and UniRef90 results of S1 and S2 Tables). Therefore, the best way to implement the SSE-PSSM is to utilize a target dataset of known structures, such as a non-redundant PDB or SCOP set. This strategy will bring another benefit: efficiency. Because PDB and SCOP databases are much smaller than the UniRef, carrying out SSP on them will significantly reduce the time cost. For instance, nrPDB90-2015 is approximately one-thousandth of UniRef90-2015 in size. As S7 Table shows, when nrPDB90-2015 was applied as the target set to run PSI-BLAST-based SSP methods, the speed was enhanced by 410 folds on average. Although using a small target set may decrease the accuracy (see also [73] for the influence of target set size on SSP accuracy), if the proposed SSE-PSSM could be implemented in those powerful methods, this decrease might be compensated or even overcome.

We designed a second-level machine learning approach to preliminarily test the accuracy of several SSP methods as if the SSE-PSSM were integrated into their feature sets. Before the test, we have demonstrated that this second-level machine learning wrap had little influence on the normal performance of the applied SSP methods (Materials and methods). Thus, accuracy improvements observed in the preliminary feature integration tests (Figs 2 and 3) were accomplished because of the integrated SSE-PSSM. Notwithstanding, this approach was just a simulation, and it had a weakness. When the output of a program was not the probabilities of SSEs but a single SSE classification, like in the Scorpion [3], there will be only one informative feature obtained for each residue, and hence the applicability of this wrapped integration may be reduced. Nevertheless, as a preliminary test, our results manifested the potential of the SSE-PSSM for serving current SSP methods as a new predictive feature set.

## Future perspectives

Even though more and more powerful machine learning technologies have been applied in predicting protein secondary structures and both the records of Q3 and Q8 are broken in every short period, the improvements accomplished recently were minor. A prevailing opinion about the decelerated improvement in SSP is that the Q3 of recent works has been approaching the 88–90% theoretical limit [14, 70]. However, this limit was just an estimate, and Rost *et al*. had also reported a large standard deviation associated with it (88.8±9.1%; Fig 2 of [70]). If we were not so closely approaching the ultimate limit of SSP (refer to [78] for stringent evaluation of the current SSP methodology where SSE-PSSM applied), there must be certain factors restricting the progress. Since the amino acid PSSM was introduced into this field in around 2000, there have been few fundamental changes in the predictive feature set. In order to push the accuracy of current SSP algorithms out of the plateau, perhaps it is time to refine the foundation of the entire SSP methodology. The proposed SSE-PSSM may serve as a good alternative or addition to the current AA-PSSM feature set and help make a breakthrough.

The idea of SSP by synthesizing the secondary structure information of the query protein's homologs had been tested. The PROSP algorithm of the HYPROSP approach (a hybrid pipeline of PROSP and PSIPRED) [79] used PSI-BLAST to retrieve structural homologs of the query and analyzed their alignments by statistics to construct a knowledge base, which contained oligopeptide fragments with associated structural information, for making predictions. The Q3 of PROSP was 60–80%, depending on the match rate of the knowledge base with the query [79]. The knowledge base is analogous to SSE-PSSM, in which every residue of the query is assigned with a set of SSE scores. Besides the difference that PROSP encoded secondary structures in three-state while the SSE-PSSM did in eight-state, PROSP made predictions by assigning the SSE of the highest score to each residue, whereas SSE-PSSM was proposed to act as a standard PSSM to be processed by machine learning. MUPRED, which encoded the alignments between the query protein and its structural homologs with a fuzzy *k*-nearest neighbor algorithm into fuzzy three-state SSE class memberships for every residue, integrated the class memberships with PSI-BLAST PSSM into a feature set to make SSP by a neural network and achieved 79–80% Q3 [80]. SSE-PSSM took advantage of the eight-state SSE classification and could encode the secondary structure of homologs in more detail than did the fuzzy class memberships. Regardless of some algorithmic differences, the concepts of SSE knowledge base and class memberships were similar to SSE-PSSM. If those scoring schemes could be assessed with current machine learning techniques, their accuracy would be remarkably enhanced. To facilitate the evolution of SSE-based SSP concepts and the renewal of modern SSP feature sets, we have implemented the proposed algorithm into an SSE-PSSM

generating web server and standalone programs available at http://10.life.nctu.edu.tw/SSE-PSSM.

SSP is an essential basis for many research and applications. If future SSP methods can adopt SSE-PSSM, a fundamental advancement in SSP is expectable and supposed beneficial for a variety of fields. For example, a homology modelling system for protein complexes we developed integrates predicted secondary structures into the template search algorithm [81]. Introducing an accurate SSP engine into this system may help improve the quality of identified templates and enable it to predict the contact interface of subunits. In bioengineering, predicting viable circular permutation sites is restricted to proteins with known structures at present because many predictive features were derived from secondary and tertiary protein structures [40, 42]; with accurate SSP, predicting viable permutation sites for proteins with only sequences will be achievable and thus expanding the application of circular permutation.

## Materials and methods

### Experimental datasets

When performing sequence similarity searches to generate PSSMs, one needs a set of query sequences and a set of target sequences. Evaluating the performance of a machine-learning-based predictor requires a set of training data and a set of testing data. If the composition of the testing dataset is very different from that of the training dataset, the extent of information leakage is supposed low, and overestimation of the predictor may be prevented. For protein sequences, the difference between datasets is typically judged by their non-redundancy of sequence identities. Following previous studies, we termed a testing dataset very different from the training dataset, *i.e.*, sharing <25% sequence identities, as an independent dataset. The primary purpose of data preparation in this study was to sustain stringent evaluation of the proposed algorithm by independent tests, and it was achieved by 1) creating a highly non-redundant pair of query datasets (QuerySet-T and QuerySet-I) for training and independent tests, and 2) utilizing well-developed independent test datasets from previous studies.

**Datasets for training and independent tests.** The experimental structural data were obtained from the 30% identity non-redundant (nr) dataset released by PDB [75] in Dec. 2015 and then homology-reduced to 25% sequence identity. Proteins with chain breaks or shorter than 20 residues were discarded. Since the official PDB nr sets were prepared by heuristic clustering [82], for ensuring the <25% identity, homology reduction methods CD-HIT [83], USEARCH [84], and MMseqs2 [85] were iteratively applied to it until no sequence could be removed at the 25% identity cutoff. In the produced dataset, abbreviated nrPDB25-2015 (11,449 proteins; S1 File), any two sequences were very different; when a sequence was selected from it to be the query protein and the rest were used as the target dataset, all the hits retrieved by sequence search would be very different from the query. To evaluate the performance of the SSE-PSSM feature set on proteins of different structural classes and sizes, we randomly selected 90 small (<150 residues), 90 medium (150–299 residues), and 90 large proteins (≥300 residues) from each of the four structural classes: all-alpha (a), all-beta (b), alpha-beta mixed (a/b) and alpha-beta segregated (a+b) according to SCOP [74]. There were 12 categories (4 classes × 3 sizes) comprising 1,080 (90 × 12) proteins. Form each category, 60 proteins were randomly selected to constitute the query dataset for training (QuerySet-T), and the other 30 were collected into the query dataset for independent tests (QuerySet-I). Finally, the QuerySet-T and QuerySet-I contained 720 and 360 proteins, respectively. After removing the sequences appearing in these query sets, the rest of nrPDB25-2015 served as the target dataset, which shared <25% identities with the query sets and was abbreviated TargetSet-nr25. Because of the stringent homology reduction, the number of sequences available for assessing the SSE-PSSM

was small. In order to ensure the results are stable, random sampling was applied three times to generate the QuerySet-T, QuerySet-I, and TargetSet-nr25 datasets, meaning all experiments in this study were repeated three times, and the reported data were average.

**Datasets from previous studies.** To make rigorous assessments of the proposed method, we sought for suitable independent test datasets. In a review by Dr. Yaoqi Zhou, state-of-the-art SSP techniques developed **before** 2016 were evaluated with two independent datasets comprising structures released **after** Jan. 1$^{st}$ 2016 with very low sequence identities to those released before 2016 [14]. These two datasets, the CASP12 obtained from the 12$^{th}$ biannual meeting of Critical Assessment of Structure Prediction techniques [67] (15 proteins of 91–540 residues), and the TS115 prepared by them (115 proteins of 43–1,085 residues), were utilized because the predictive model of SSE-PSSM and most SSP algorithms tested in this study were trained with proteins released before 2016. Since some tested SSP algorithms were released after 2016, the CASP13 obtained from the 13$^{th}$ biannual meeting of CASP consisting of structures released between 2017 and 2019 was also utilized (43 proteins of 32–718 residues). When performing independent tests with these query sets, the target datasets for generating PSSMs were the UniRef90 of 2015 (UniRef90-2015; 38.2 million proteins) [86] and a 90% identity nr set of PDB of the same year (nrPDB90-2015; 34,835 proteins; see S2 File). Most SSP methods applied in this study were template-free algorithms. We also tested the template-based version of SSpro8 (SSpro8T) to compare different categories of methods. The SSpro8T was evaluated by a large testing dataset of ~11,000 proteins, which was not released [6]. Following the preparation procedure stated in [6], we established an equivalent testing dataset containing 10,226 proteins abbreviated as TSsspro8 (listed in S3 File with the CASP13).

## Applied secondary structure prediction methods

**Methods using PSI-BLAST as the PSSM generator.** Several PSI-BLAST-powered SSP methods were applied to help evaluate the SSE-PSSM and determine whether integrating this new feature set into those advanced methods would improve SSP accuracy. The standalone programs of all those methods were trained and released before 2016, including three-state algorithms: PSIPRED (v3.3) [1], SpineX (v2.0) [2], Scorpion (v1.0) [3], and Spider2 (v2.0) [4], and eight-state algorithms: RaptorX (v1.0) [5], SSpro8 (v5.2) [6], and DeepCNF (v1.02) [7]. The original packages of these programs used different versions and parameters of PSI-BLAST, such as the E-value cutoff and the number of iterations of sequence similarity searches. To make the experiments performed on an equal basis, we modified their pipeline scripts such that the psiblast program of NCBI blast 2.3.0 [53] with a given set of parameters became the only PSSM engine (see S8 Table for the original and modified settings). The way we set up PSI-BLAST parameters was based on 1) the common settings used by most applied methods, 2) the settings suggested by the secondary structural code generating algorithms utilized in this work [77, 87], and 3) the default of psiblast.

To make sure that we correctly ran these programs and our modifications did not disturb their normal performance, as a pretest, we conducted SSP using these methods with the TS115 and CASP12 query sets and the UniRef90-2015 target dataset and compared the results with previous reports on the same methods using equivalent datasets [7, 14]. In most, if not all, previous SSP studies, the accuracy data were presented as average values over queries. The averages of accuracy computed in this way using our pipeline were all close to and even slightly higher than those obtained from the literature (Table 4). In this experiment, we also presented the accuracy measures as their average over residues. These accuracy values were all close to but a little lower than the values averaged over queries. The average of a measure Q over

**Table 4. Results of performance pretest for several state-of-the-art SSP methods.**

| SSE set | Method | Datasets: TS115 vs UniRef90-2015 | | | Dataset: CASP12 vs UniRef90-2015 | | | Dataset: CASP13 vs UniRef90-2015 | |
|---|---|---|---|---|---|---|---|---|---|
| Three-state | Measure | Q3, literature | Q3, avg over queries[c] | Q3, avg over residues[c] | Q3, literature | Q3, avg over queries[c] | Q3, avg over residues[c] | Q3, avg over queries | Q3, avg over residues |
| | Scorpion[a] | 0.817 | 0.819 (0.002) | 0.816 (-0.001) | 0.805 | 0.821 (0.016) | 0.822 (0.017) | 0.793 | 0.796 |
| | Spider2[a] | 0.819 | 0.823 (0.004) | 0.819 (0.000) | 0.798 | 0.806 (0.008) | 0.813 (0.015) | 0.802 | 0.801 |
| | SpineX[a] | 0.801 | 0.805 (0.004) | 0.801 (0.000) | 0.769 | 0.779 (0.010) | 0.783 (0.014) | 0.776 | 0.777 |
| | PSIPRED[a] | 0.802 | 0.811 (0.009) | 0.800 (-0.002) | 0.780 | 0.780 (0.000) | 0.781 (0.001) | 0.782 | 0.781 |
| | DeepCNF[a] | 0.823 | 0.826 (0.003) | 0.819 (-0.004) | 0.821 | 0.819 (-0.002) | 0.829 (0.008) | 0.798 | 0.800 |
| | RaptorX[b] | 0.812 | 0.813 (0.001) | 0.808 (-0.004) | 0.791 | 0.790 (-0.001) | 0.794 (0.003) | 0.778 | 0.779 |
| | SSpro8[b] | 0.795 | 0.798 (0.003) | 0.789 (-0.006) | 0.776 | 0.771 (-0.005) | 0.778 (0.002) | 0.762 | 0.766 |
| | *Average* | *0.810* | *0.814 (0.004)* | *0.807 (-0.003)* | *0.791* | *0.795 (0.004)* | *0.800 (0.009)* | *0.784* | *0.786* |
| Eight-state | Measure | Q8, literature | Q8, avg over queries[c] | Q8, avg over residues[c] | Q8, literature | Q8, avg over queries[c] | Q8, avg over residues[c] | Q8, avg over queries | Q8, avg over residues |
| | DeepCNF[a] | 0.720 | 0.716 (-0.004) | 0.703 (-0.017) | 0.730 | 0.714 (-0.016) | 0.728 (-0.002) | 0.681 | 0.669 |
| | RaptorX[b] | 0.697 | 0.709 (0.012) | 0.699 (0.002) | 0.651 | 0.680 (0.029) | 0.693 (0.042) | 0.670 | 0.660 |
| | SSpro8[a] | 0.680 | 0.686 (0.006) | 0.672 (-0.008) | 0.690 | 0.664 (-0.026) | 0.675 (-0.015) | 0.642 | 0.635 |
| | *Average* | *0.694* | *0.704 (0.010)* | *0.691 (-0.003)* | *0.679* | *0.686 (0.007)* | *0.699 (0.020)* | *0.664* | *0.655* |

[a]Obtained from [14], where the query datasets were TS115 and CASP12, and the target dataset was UniRef90-2015.

[b]Obtained from [7], where the query datasets were CullPDB and CASP11, and the target dataset was UniRef90-2015.

[c]Values in parentheses are the differences between the measures computed here and those obtained from the literature.

queries ($\overline{Q_{qry}}$) and residues ($\overline{Q_{res}}$) can be expressed as,

$$\overline{Q_{qry}} = \frac{\sum_{q=1}^{n} Q(q)}{n} = \frac{\sum_{q=1}^{n} \frac{N_r(q, accurate)}{N_r(q)}}{n} \tag{1}$$

$$\overline{Q_{res}} = \frac{\sum_{q=1}^{n} N_r(q, accurate)}{\sum_{q=1}^{n} N_r(q)} \tag{2}$$

where $q$ stands for a query protein, $n$ is the number of query proteins, $Q(q)$ denotes the $Q$ accuracy of $q$, $N_r(q)$ represents the number of residues of $q$, and $N_r(q,accurate)$ means the number of residues of $q$ that are accurately predicted. The $\overline{Q_{qry}}$ computes the proportion of correct predictions for each query and makes the arithmetic mean over all queries. It weights **proteins** equally, regardless of their sizes, thus tending to overweight small ones. The $\overline{Q_{res}}$ divides the total number of correctly-predicted residues by the total number of residues. It weights all **residues** equally and reduces the effects of the size imbalance of query proteins. For example, if the Q3 of a 50-residue protein A is 0.90 and that of a 350-residue protein B is 0.50, then the $\overline{Q_{qry}}$ of A and B will be 0.70. However, among the 400 (= 50 + 350) residues of A and B, only 220 (= 50×0.90 + 250×0.50) are correctly predicted. Using the same example, the $\overline{Q_{res}}$ will be 0.55 (= 220 / 400), clearly reflecting the percentage of correct predictions among all residues. In fact, $\overline{Q_{qry}}$ and $\overline{Q_{res}}$ are respectively analogous to the macro- and micro-average commonly used to assess machine-learning classifiers. In a multi-class system, micro-average is preferable because of its robustness in the face of the imbalance of training data. Since query sets with different distributions of protein sizes were utilized in this study, even though the $\overline{Q_{res}}$ values

were slightly lower than the $\overline{Q_{qry}}$ in this pretest, we decided to use this micro-averaged accuracy for all experiments.

**Methods using HHBlits as the PSSM generator.** HHBlits is an efficient iterative protein sequence similarity search algorithm that aligns sequences according to their hidden Markov models [68]. Since the MSA conservation profile of HHBlits can be readily transformed into a PSSM, it is increasingly utilized by recent SSP methods as a PSSM generator. The HHBlits-based SSP algorithms applied in this study fall into two types. The NetSurfP-2 [10] used only HHBlits to generate its SSP feature set, while the others used both the PSSMs generated by HHBlits and PSI-BLAST as predictive features, including MUFOLD-SS [9], Porter5 [11], and Spider3 [8]. As listed in S8 Table, we also unified the version and settings of PSI-BLAST used in the pipelines of these programs. Additionally, the version of HHBlits was fixed to be 3.3.0. These HHBlits-based SSP algorithms were significantly more accurate than the PSI-BLAST-based ones. Assessed with the TS115 and CASP12/13 independent datasets, both Q3 and Q8 of the former were >3% higher than the latter (see S5 Table).

## Algorithm of the SSE-PSSM

The major innovation of this work is the secondary structure element-based PSSM, the algorithm of which is illustrated in Fig 4. Most of the steps follow the traditional procedure for constructing an amino acid-based PSSM. The key modification is an SSE transformation.

1. Sequence similarity search. Take one query sequence, use a search tool like PSI-BLAST or HHBlits to search against a given target set, retrieving a hit list of homologous sequences (Fig 4A). If PSI-BLAST is used, make sure it provides the alignment between the query and each hit by setting the same value for "num_descriptions" and "num_alignments" parameters (S8 Table). If HHBlits is used, the multiple sequence alignment between the query and all hits can be obtained from its .hhr output file. For ensuring that the produced PSSM is not biased, in a careful experimental design, the hits should be filtered according to sequence homology [53]. We did not need to do it because all the hits had been highly non-redundant from the query (<25% identity; see **"Experimental datasets"**).

2. Obtaining the SSE sequences of the hits (Fig 4B). For each hit protein $h$, if its structure is available, use DSSP [49] to compute its SSE sequence. Otherwise, make a representative SSE sequence of it by 1) using PSI-BLAST/HHBlits to find its homologs from a reference structure dataset, 2) computing the SSE sequences of the secondary homologs identified from the reference dataset by DSSP, and 3) for each residue $r$ of $h$, based on the alignment between $h$ and its homologs, collecting the SSE codes of all residues aligned with $r$ and determining the representative SSE code by vote.

3. SSE transformation (Fig 4A). After obtaining all SSE sequences of the hits, for each hit, based on its alignment with the query, replace per residue the amino acid code with the SSE code. For instance, if the amino acid and SSE sequences of a hit are "`ILGWL`" and "`CHHEE`", respectively, and the way the hit aligned with the query is "`IL-GW--L`", then the transformed alignment string will be "`CH-HE--E`". The result of this step is a set of sequences, the residues of which were aligned according to their amino acid sequences, but the apparent residue codes were their SSE.

4. Computation of the code occurrences. For each residue position $p$ of the query, according to the SSE-transformed alignments, compute the occurrences (*Occ*) of each SSE code

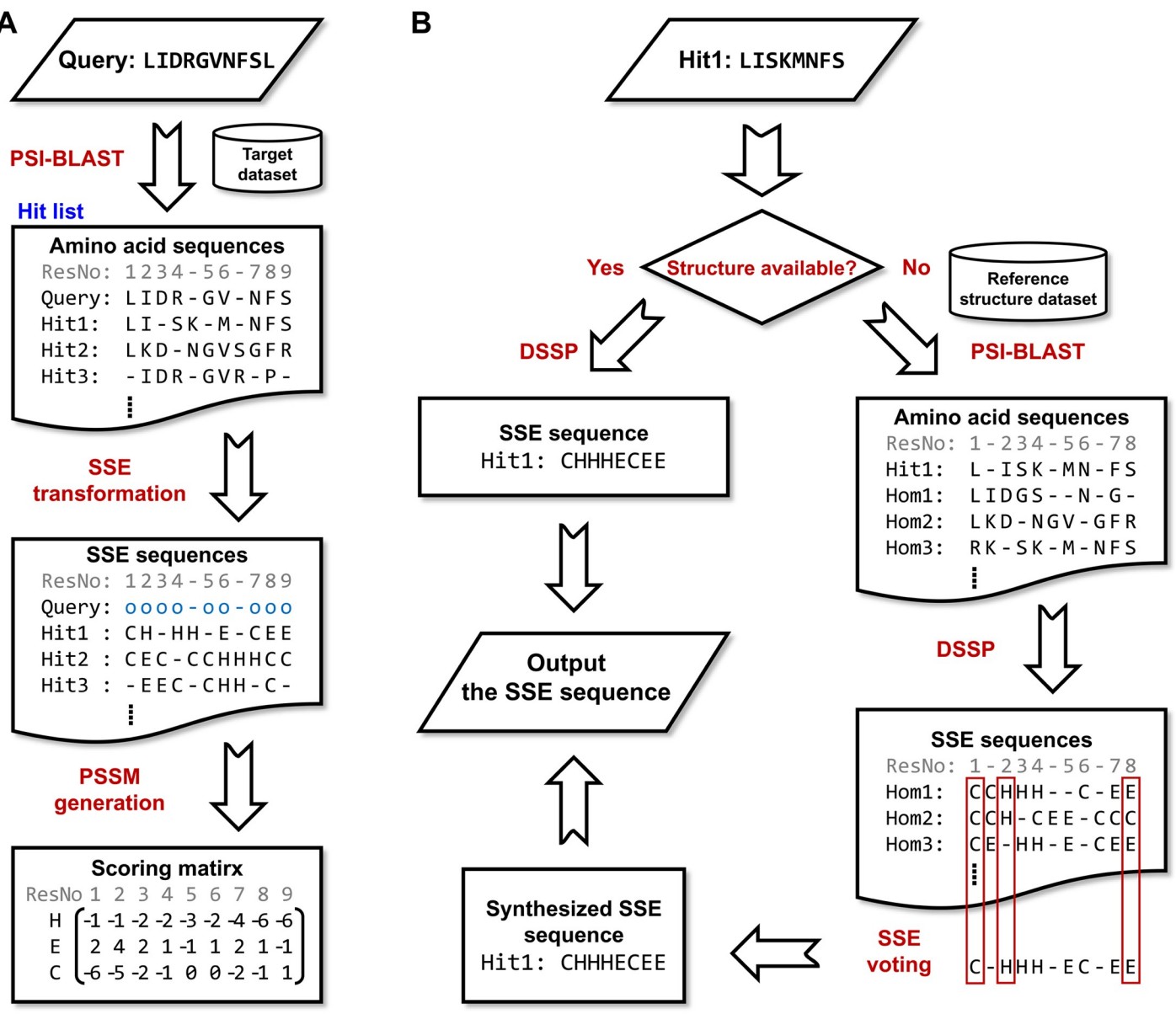

**Fig 4. Flowchart of the SSE-PSSM algorithm.** (A) The core procedure of the algorithm. (B) Determination of the SSE sequence of a hit. A sequence similarity search is performed to retrieve a hit list for the query sequence. Next, for each hit protein, directly obtain its SSE sequence from the known structure or synthesize one by position-specific voting according to the homologs (Hom) of the hit retrieved by the second-round sequence similarity search against a reference protein structure dataset. An SSE transformation of the sequence alignments between the query and hits is then carried out by replacing the amino acids with SSE codes. Finally, the PSSM is generated according to the transformed alignments.

among those residues aligned with $p$ using equations,

$$Occ(c, p) = \sum_{h=1}^{n_h} I(Code_{h,p}, c) \times W(h) \quad (3.1)$$

$$I(x, c) = \begin{cases} 1, & x = c \\ 0, & otherwise \end{cases} \quad (3.2)$$

where $c$ stands for a code symbol, $n_h$ denotes the number of hits, $Code_{h,p}$ means the code symbol of the residue of hit $h$ aligned with $p$, and $W(h)$ is a weight function of $h$. The purpose of a weight function is to make the hits that may carry more information contribute more to the matrix [53]. For the speed and simplicity of implementation, the weight function we used was the sequence similarity between the query protein and $h$.

5. Generation of the position propensity matrix (PPM). Based on the occurrence data, a PPM can be created using Eq (4) (see Fig 1 for an illustration).

$$PPM_{s,p} = \frac{Occ(s,p)}{\sum_{c=1}^{n_s} Occ(c,p)} \tag{4}$$

where $s$ is the set of code symbols in the SSE alphabet, $n_s$ denotes the number of symbols, and $c$ stands for a code symbol. Several SSE alphabets were tested. See **Settings of the principle factors** and S9 Table for details.

6. Conversion of the PPM to the final PSSM. The elements in the proposed SSE-PSSM are calculated as log likelihoods according to the BLOSUM algorithm [88] using the formula,

$$PSSM_{s,p} = \log_2(PPM_{s,p}/B_s) \tag{5}$$

where $B_s$ is a collection of the background occurrence frequencies of each SSE code symbol in the code set $s$. The background frequencies were computed based on the reference dataset.

7. If there is no hit retrieved for the query sequence in Step 1 or there is no aligned residue from the hits for any residue of the query, the pseudocount method [53] is applied to compute the substitution scores. The prerequisite of the pseudocount method is a pre-established substitution matrix of SSE codes. The matrix used in this study was produced based on nrPDB-2015 using SARST structure alignments and the BLOSUM algorithm [77]. See S4 File for the matrix and the details of its production.

## Computation of the classic secondary structure prediction features

**Amino acid-based PSSM.** Following most previous studies [1–7], the AA-PSSM was generated by the PSI-BLAST [53]. In the experiment of Fig 3, it was also generated by HHBlits [68]. For every residue of the query, this PSSM provides 20 substitution scores for training the predictive model by machine learning.

**Amino acid type.** Many recent studies also used each residue's amino acid type as a feature [7, 89, 90]. For instance, Dr. Wang and colleagues used 21 binary features to indicate the amino acid type (20 types plus the undetermined) at a residue position as they developed the DeepCNF [7]. Here we proposed a strategy to encode amino acid types as a feature: 1) classify amino acids into five classes based on their side-chain physiochemical properties [91], 2) sort the class by hydrophobicity high to low, 3) sort the amino acids in each class by their natural abundance low to high, and 4) assign an integer, low to high, to the sorted amino acids in all classes (see S5 File for the encoded amino acids). The benefits of this strategy include the speed of machine learning and implicated biological meaning. First, the proposed strategy encodes 20 amino acids into just one feature, helpful for keeping the feature set small. Second, we have tested several other methods to encode the amino acids, inclusive of the DeepCNF algorithm, the molecular weight, the hydrophobicity scale, the side chain hydropathy, the solvent accessible surface area, the isoelectric point, the radius of gyration of the side chain, and the

occurrence frequency in the PDB. Most of them produced similar SSP accuracies, and the proposed amino-acid feature slightly outperformed them. We presumed that the proposed classification step resulted in the fact that physiochemically similar amino acids were assigned with similar integers, making it easier for machine learning algorithms to identify the relationship or closeness between amino acids.

**Integration of the SSE-PSSM with standard features.** There were three major types of predictive features used in this study, the SSE-PSSM, the amino acid PSSM, and the amino acid type. The integration of any two or all of them was simply a merge of the feature sets. Since the amino acid type is usually applied in recent works [7, 89, 90], the final version of the SSE-PSSM assessed in this study was integrated with it, so was the amino acid PSSM.

## Application of machine-learning methods

We did not develop any new machine learning method but utilized a system we had to evaluate the SSE-PSSM as a set of SSP features. If the advanced machine learning methods of recent studies, such as the deep convolutional neural fields (CNF) [7], bidirectional recursive neural networks (BRNN) [6, 57, 65, 66], and convolutional recurrent neural network (CRNN) [90, 92] can be applied with the SSE-PSSM, the outcome will be much better than the reported.

Previously as we studied a protein structural rearrangement phenomenon known as the circular permutation [87], an artificial intelligence system was developed to integrate several machine learning, random sampling, and parameter optimization algorithms [41, 42]. In the present work, this system was applied, and the recruited algorithms included bootstrap sampling, decision tree, and artificial neural network. After obtaining the answers and feature values from the training query set (QuerySet-T) and TargetSet-nr25, 100 bootstrap samples each with a bootstrapped feature set were made to train minor models of decision tree and artificial neural network (50 samples for each algorithm). The final prediction model was then formed by collecting the minor models, which made predictions by vote (note that the collection of bootstrapped decision trees is known as a random forest). With this final model, the probabilities of candidate answers for a given residue could be estimated as the percentages of votes the answers received.

With the same input data and machine learning layout, either a three-state or an eight-state predictive model could be produced. An eight-state model could be applied to perform three-state predictions simply by reducing the output SSE codes to three states. However, our pre-tests showed that the accuracy of such manipulation was not comparable to that of a predictive model specifically constructed for three-state predictions. In this study, the three- and eight-state prediction experiments were done using different predictive models.

## Settings of the principle factors

**Window size.** The window spanning technique is commonly implemented in SSP, and previous studies had used various window sizes ranging from a couple of residues [40] to tens of residues [6]. A small window may delicately compile information of secondary structures with short units [3, 50], while a large window may help detect long-range residue interactions [6, 66, 93]. We tested the SSP performance of SSE-PSSM with single- to 21-residue windows. As shown in S1 Fig, the accuracy increased rapidly as the window extended from 1 to 3 residues; then, the performance seemed to reach a plateau. The highest Q3 and Q8 occurred at 6–7 residues and the highest SOV3 and SOV8 at 5 residues. Since SOV is more critical than the Q accuracy and extending the window by one residue will increase the number of features by one fold, which will greatly increase the time cost of machine learning, we decided to use the window size of 5 residues.

**SSE alphabets and their combinations.** In addition to the commonly used three-state and the DSSP-defined eight-state SSE codes, the kappa-alpha and SARST alphabets [76, 77] were examined for feasibility in constructing the SSE-PSSM feature set. Three-state prediction experiments on feature sets derived from the SSE-PSSMs of these alphabets and several combinations were performed using QuerySet-T and TargetSet-nr25 with a 5-residue window. Results listed in S9 Table indicated that combining the PSSMs of DSSP, kappa-alpha, and SARST codes might achieve the best accuracy, but the enhancement over the SSE-PSSM of DSSP codes was not much (0.4% in Q3). Considering the simplicity for future implementations of the SSE-PSSM, we decided to use only the PSSM of eight-state DSSP codes in this study.

## Combination of the SSE-PSSM with well-developed secondary structure prediction programs

The source codes of model training of the applied SSP programs were not available. In order to integrate the SSE-PSSM into their systems, an indirect strategy was applied. First, after inputting a query protein to a given SSP program, for each residue, obtain the predicted probabilities of SSE codes. According to the category of the method, there may be 3 or 8 probabilities. Second, if the output of the program is just one predicted SSE code, set the probability to be 1 for the code and assign 0 probability to the others. Third, the obtained probability values are used as features combined with the SSE-PSSM for training predictive models and performing predictions by our machine learning system. In this way, the SSE-PSSM could be "indirectly" incorporated into a compiled SSP program. This strategy had been used previously. For example, the template-based SSP method PROTEUS [94] integrated the output of PSIPRED [1], JNET [54], and its TRANSSEC into a machine learning feature set to make predictions by a neural network. If there were homologous structural fragments (*i.e.*, templates) identified for the query sequence, PROTEUS would overlap the structural fragments over the predicted secondary structure to be its final output. With its template mode disabled, the Q3 of PROTEUS was 79.4%, which reasonably preserved and integrated the accuracy of applied programs: PSIPRED 78.1%, JNET 73.2%, and TRANSSEC 70.3% [94]. To verify whether our indirect feature-integration strategy also worked reasonably, we compared the accuracy of the SSP methods executed solely and their accuracy after the second-level machine learning wrap. S10 Table shows that the differences of accuracy between the native programs and the machine learning wrapped pipelines were generally small.

## Supporting information

**S1 Fig. Performance of different window sizes.**
(PDF)

**S1 File. The nrPDB25-2015 dataset.**
(FASTA)

**S2 File. The nrPDB90-2015 dataset.**
(FASTA)

**S3 File. The CASP13 and TSsspro8 independent test datasets.**
(XLSX)

**S4 File. BLOSUM25-SSE: The substitution matrix of SSEs generated based on structural homologs of ≥25% sequence identity.**
(PDF)

**S5 File. The encoded amino acids.**
(TXT)

**S1 Table. Performance of the combined feature set of amino acid and SSE PSSMs.**
(XLSX)

**S2 Table. Performance of preliminary incorporation of SSE-PSSM into state-of-the-art SSP methods using PSI-BLAST to generate PSSM.**
(XLSX)

**S3 Table. Performance of preliminary incorporation of the traditional amino acid PSSM into several state-of-the-art SSP methods.**
(XLSX)

**S4 Table. Performance of preliminary incorporation of SSE-PSSM into state-of-the-art SSP methods using HHBlits to generate PSSM.**
(XLSX)

**S5 Table. Performance pretests for the state-of-the-art SSP methods utilized in this study using different PSSM generating engines and independent test datasets.**
(XLSX)

**S6 Table. Influence of the proportion of low or null occurrence codes in position propensity matrix on the SSP accuracy of several residue alphabets.**
(PDF)

**S7 Table. The computation time of several SSP methods running on target datasets with a large difference in size.**
(XLSX)

**S8 Table. PSI-BLAST settings of the applied SSP methods.**
(XLSX)

**S9 Table. Accuracy of feature sets comprising different combinations of SSE-PSSMs.**
(PDF)

**S10 Table. Accuracy of state-of-the-art SSP methods running in indirect combination with the SSE-PSSM features.**
(PDF)

## Acknowledgments

We would like to thank Chia-Hua Lo and Chia-Tzu Ho, students of WCL, for their great help in data analyses and preparation of the Supplemental Information files. This study was greatly accelerated owing to the computing power offered by Prof. Jinn-Moon Yang and Jenn-Kang Hwang at National Yang Ming Chiao Tung University and Prof. Ping-Chiang Lyu at National Tsing Hua University, Taiwan.

## Author Contributions

**Conceptualization:** Wei-Cheng Lo.

**Data curation:** Teng-Ruei Chen, Sheng-Hung Juan, Yu-Wei Huang, Yen-Cheng Lin, Wei-Cheng Lo.

**Funding acquisition:** Wei-Cheng Lo.

**Investigation:** Teng-Ruei Chen, Sheng-Hung Juan, Wei-Cheng Lo.

**Methodology:** Sheng-Hung Juan, Wei-Cheng Lo.

**Project administration:** Wei-Cheng Lo.

**Resources:** Yu-Wei Huang, Wei-Cheng Lo.

**Software:** Teng-Ruei Chen, Sheng-Hung Juan, Yu-Wei Huang, Yen-Cheng Lin.

**Supervision:** Wei-Cheng Lo.

**Validation:** Teng-Ruei Chen, Sheng-Hung Juan, Yu-Wei Huang, Yen-Cheng Lin, Wei-Cheng Lo.

**Visualization:** Yu-Wei Huang.

**Writing – original draft:** Teng-Ruei Chen, Wei-Cheng Lo.

**Writing – review & editing:** Sheng-Hung Juan, Yu-Wei Huang, Yen-Cheng Lin, Wei-Cheng Lo.

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
