## [Decision Letter · Decision Letter 0]

30 Sep 2020

PONE-D-20-15212

A secondary structure-based position-specific scoring matrix applied to the improvement in protein secondary structure prediction

PLOS ONE

Dear Dr. Lo,

Thank you for submitting your manuscript to PLOS ONE. After careful consideration, we feel that it has merit but does not fully meet PLOS ONE’s publication criteria as it currently stands. Therefore, we invite you to submit a revised version of the manuscript that addresses the points raised during the review process.

An important point will be to shorten the manuscript and address the multiple questions raised by the reviewers. Please also underline the specificity of your approach in regards to recent PLoS ONE publication (https://journals.plos.org/plosone/article?id=10.1371/journal.pone.0235153#sec013)

We look forward to receiving your revised manuscript.

Kind regards,

Alexandre G. de Brevern, Ph.D.

Academic Editor

PLOS ONE

Journal Requirements:

Reviewers' comments:

Reviewer's Responses to Questions

**Comments to the Author**

1. Is the manuscript technically sound, and do the data support the conclusions?

Reviewer #1: Partly

Reviewer #2: Yes

2. Has the statistical analysis been performed appropriately and rigorously? 

Reviewer #1: N/A

Reviewer #2: Yes

3. Have the authors made all data underlying the findings in their manuscript fully available?

Reviewer #1: Yes

Reviewer #2: Yes

4. Is the manuscript presented in an intelligible fashion and written in standard English?

Reviewer #1: Yes

Reviewer #2: Yes

5. Review Comments to the Author

Reviewer #1: The work by Chen et al proposed a method to generate secondary structure PSSM based on homologous sequences in a predefined library and then use this PSSM for secondary structure prediction. The work is interesting. Here are some questions.

1) The literature review did not cover many methods that incorporate homology in secondary structure prediction. Please compare and contrast the difference. Examples are HYPROSP, PROTEUS, and MUPRED.

2) It is not clear what parameters were used in PSI-BLAST to locate the homologous sequences of a query sequence from the target dataset. Was a e-value cutoff used? What happens if no homologs were found in the target dataset?

3) Please report the separate results of free-modeling targets in CASP12 and CASP13.

4) For completeness, please compare to more recent works of SPOT-1D (Bioinformatics. 2019 Jul 15;35(14):2403-2410) and NetSurfP-2.0 (Proteins. 2019 Jun;87(6):520-527).

5) This method essentially is a template-based method (remote-homology). Please explicitly say so.

Reviewer #2: The authors have presented a secondary structure prediction algorithm using a machine learning model trained on PDB structures with DSSP SSE assignments.

The mapping of secondary structure information on pairwise sequence alignment to generate a position specific scoring matrix is very simple to understand and is presented in an intelligible fashion.

Although, the use of local pairwise alignment for structure template based algorithms is very risky and can prove to be fatal yet almost every other template based sequence to structure predictor does exactly that.

Therefore, I shall not comment over such fundamental ideological discrepancies amongst sequence and structural specialists.

Overall, the article is well explained and does appear to be lengthy to read and comprehend. This is a major drawback in the manuscript organization.

I would like to suggest the authors to cut-short the length of the article for convenient reading. Longer articles tend to have lesser citations.

Before this article can be accepted to be published, I have some major concerns in the designed study and reported statistics. Concerns are listed as follows:

Major_1: In Table 6 the authors have tried to show a benchmark (control) analysis by running the known SoA SSP methods over standardized data-set after tinkering their protocol with PSI-BLAST v2.3.0.

1) Authors had used a version of 2015 and not a recent one. While I would like to know the rationale behind the choice, my important concern is that:

The references listed as no. 6 and 7 were published in 2014 and 2016 resp. and used the same or perhaps an older version of PSI-BLAST.

The NCBI-BLAST package updates from v2.2.28 to v2.3.0 does not list any major change to PSI-BLAST pipeline.

Did the improvements made to HSP calculation and/or handling of compositional biases in BLAST, impacted the difference between the performances of SSpro8, DeepCNF as listed in the references and as reported in this study?

2) The authors have deliberately tried to cite SSPro8's performance from listed ref. 7(DeepCNF's Sci. Reports, 2016) and DeepCNF's accuracy from listed ref. 10 (2018 review) rather than using the values from their own studies [listed ref. 6 and 7].

If the authors used the accuracy values from SSPro8's and DeepCNF references then their accuracy should be 87.92% and 85.40% instead of 79.5% and 82.3% resp.

The reported values for SSPro8 are also misleading as the article reports the SSPro8 values for older template free method while SSPro8 published in 2014 showcases drastic improvements in prediction when employing secondary structure data.

Since this article is specifically about a structure based PSSM, therefore, I expect a clarification on the choices that the authors made to report these values.

3) The listed ref 6 for SSpro8 claims the secondary structure prediction problem to be essentially solved by showcasing a 92% accuracy using their structure template based approach.

While their claim can be disputed, it would be worthwhile to check the performance of SSE-PSSM on the exact same data-set as used in ref. 6.

Major_2: Authors have not disclosed the quality of PDB structures used. Since, the PSSM is based on DSSP secondary structure assignment, it is imperative for the readers to know:

1. What was the resolution cut-off?

2. Were there any missing regions in the final PDB data-set used, if so, how was it handled?

Major_3: It is quite intriguing to understand that what motivated the authors to use three and not one sequence homology based clustering approaches to reduce the data-set size.

Currently, CD-HIT, USEARCH, and MMseq2 were used to generate nrPDB25-2015 data-set.

Minor_1:

Reduce the manuscript size by making good use of the supplementary space.

1. Transform data Tables to data-point plots with important data labels visible. The crude tables can be provided in the supplementary files for the curious minded.

2. Lots of redundant sentences can be found in the main text. Please remove these.

Ex:

Ln.560-563 and 564- "The experimental protein structural data were obtained from the 30% identity non-redundant dataset released by the Protein Data Bank [72] in 2015 and were then homology-reduced to 25% sequence identity." "All proteins in this source dataset were deposited in the PDB no later than Dec. 2015."

Ln 380-384 is completely irrelevant and not required.

Suggestion:

After reviewing the article, I strongly encourage the authors to rather develop this methodology as a web-service and compile it for a specialized journal where it shall have its due attention and criticism.

6. PLOS authors have the option to publish the peer review history of their article (what does this mean?). If published, this will include your full peer review and any attached files.

Reviewer #1: No

Reviewer #2: No

---

## [Author Response · Author response to Decision Letter 0]

23 Apr 2021

*** A Word file of our responses to Reviewers' Comments has been uploaded. Although we make a plain text version here following the instruction of the submission system of PLOS ONE, we would like to suggest viewing the Word version.

## Response letter

PONE-D-20-15212

A secondary structure-based position-specific scoring matrix applied to the improvement in protein secondary structure prediction

Dear Professors and Editor,

We would like to thank the anonymous reviewers for their careful reading and detailed comments, which have greatly enhanced this article. We are pleased that you find sufficient merit in the work and ask for an appropriately revised manuscript. The new version of our manuscript has been modified according to referees' comments, which are also answered as follows.

Editor's comment

Please also underline the specificity of your approach in regards to recent PLoS ONE publication (https://journals.plos.org/plosone/article?id=10.1371/journal.pone.0235153#sec013).

## Response:

We thank the editor for handling this manuscript. The topic of the above publication was about the speed improvement in protein secondary structure prediction (SSP). The improving strategy we proposed in that paper was based on size and homology reductions of the PSI-BLAST target dataset. There was no predicting method or machine learning feature set proposed, but these are the topic of the present work. The speed improvement paper has been cited in the revised manuscript where we find it appropriate (Ref [73]), and what motivated us to do that research is stated later in this letter.

Comments to the Author

1. Is the manuscript technically sound, and do the data support the conclusions?

Reviewer #1: Partly

Reviewer #2: Yes

## Response:

Sincerely, we would like to thank the reviewers for their constructive comments. We did our best to fulfill the requirements for publication. Please see below for our response to their concerns.

2. Has the statistical analysis been performed appropriately and rigorously?

Reviewer #1: N/A

Reviewer #2: Yes

## Response:

Indeed, complicated statistical analyses are not required to conduct this research. In calculating the average of accuracy scores, we applied the concept of micro-average in addition to the conventional macro-average used in most previous studies. This is because multiple query datasets were utilized in this work, and the micro-average is robust when the sizes of query proteins within or between datasets significantly vary. In Materials and Methods (Lines 659–663 on Page 35), we added an example of two proteins with very different sizes to demonstrate the difference between macro- and micro-averages, hoping to help the reader know how we reported the data.

3. Have the authors made all data underlying the findings in their manuscript fully available?

Reviewer #1: Yes

Reviewer #2: Yes

## Response:

Many thanks to the reviewers for their time and patience in reviewing our numerous Supplementary Information files.

4. Is the manuscript presented in an intelligible fashion and written in standard English?

Reviewer #1: Yes

Reviewer #2: Yes

## Response:

We thank the reviewers for their recognition of our writing quality. We have shortened the article following their suggestions.

5. Review Comments to the Author

Reviewer #1:

The work by Chen et al proposed a method to generate secondary structure PSSM based on homologous sequences in a predefined library and then use this PSSM for secondary structure prediction. The work is interesting. Here are some questions.

1) The literature review did not cover many methods that incorporate homology in secondary structure prediction. Please compare and contrast the difference. Examples are HYPROSP, PROTEUS, and MUPRED.

## Response:

We thank the reviewer for the positive feedback and informative comment. Before receiving this comment, we did not know those methods. After reviewing them, we felt so encouraged about the future application of SSE-PSSM because those methods had utilized similar concepts to the SSE-PSSM and achieved good accuracy in SSP. They are ideal supporting information about the feasibility of our algorithm. The PROTEUS established a neural network to combine the predicting power of three SSP methods, a strategy analogous to our feature-integration procedure described in Materials and Methods. Hence, we summarized the algorithm and results of PROTEUS in Materials and Methods on Page 45. The core concepts of HYPROSP and MUPRED are similar to the SSE-PSSM, although the SSE-PSSM encodes secondary structure information more delicately and is easier to implement. Information about these innovative SSP methods has been added to Discussion on Pages 28–29. We are very grateful to the reviewer for letting us know about these methods. Encouraged by them and because of the suggestion of Reviewer 2, we implemented a web server of SSE-PSSM to facilitate its future applications.

2) It is not clear what parameters were used in PSI-BLAST to locate the homologous sequences of a query sequence from the target dataset. Was a e-value cutoff used? What happens if no homologs were found in the target dataset?

## Response:

The reviewer is acknowledged for reminding us that we had not clearly described these necessary details for implementing the proposed SSP feature set. In the previous manuscript, we made a Supplemental Information file (the old S4 Table) to provide the PSI-BLAST parameter settings. However, the way we mentioned this file was obscure. In the revised manuscript, we organized the descriptions about the parameters of PSI-BLAST into a paragraph entitled "Methods using PSI-BLAST as the PSSM generator" (Page 33) and specifically named the E-value cutoff and the number of PSSM iterations to make the reader know what we mean by parameters. The E-value cutoff we applied for all PSI-BLAST-based SSP algorithms was 100, as listed in the revised S8 Table along with other parameter settings.

The second question is critical to the implementation. Actually, a reason we set the E-value cutoff as 100 (default of PSI-BLAST: 10) was to prevent the "no homolog" scenario. Another problematic scenario for generating a PSSM is that, for some residue position, there is no aligned residue from the homologs. In addition to the E-value cutoff, we established a second firewall for these challenging scenarios using the pseudocount method utilized by the PSI-BLAST [53]. Before this method could be applied, a substitution matrix of SSE codes based on homology analysis should be established. To make future implementation of SSE-PSSM easy, we have added S4 File to provide this matrix and how it was generated (cited in Line 741, Page 40).

3) Please report the separate results of free-modeling targets in CASP12 and CASP13.

4) For completeness, please compare to more recent works of SPOT-1D (Bioinformatics. 2019 Jul 15;35(14):2403-2410) and NetSurfP-2.0 (Proteins. 2019 Jun;87(6):520-527).

## Response:

We appreciate that the reviewer suggested testing the proposed feature set with more updated independent test datasets and SSP algorithms, which have made this study much stronger.

The SSP algorithms assessed in the original manuscript were all published before 2016 because this study began in 2016. Due to the limited computation resources of our laboratory and the heavy loading of some applied algorithms, every experiment was so time-consuming that finishing the first manuscript took us three years. When this study was initiated, most SSP methods used PSI-BLAST to generate PSSM. Recently, more and more algorithms have begun using both the PSSMs generated by PSI-BLAST and HHBlits (e.g., SPOT-1D, MUFOLD-SS, Porter5, and Spider3), or only the HHBlits PSSM, such as NetSurfP-2. 

Between the recommended methods, we have applied the NetSurfP-2. SPOT-1D was successfully installed. However, predicting one protein of ~200 residues with it using our rapidest server machine (16-thread Intel i9 5.1GHz CPU, Nvidia GTX 1280-core 1.7GHz GPU, and 768G RAM) took 9.6 days. Due to the limited time and GPU resources, we turn to use other recent outstanding algorithms, including MUFOLD-SS, Porter5, and Spider3, to make assessments. In particular, Spider3 was published a little earlier than SPOT-1D by the same team. SPOT-1D makes predictions by integrating the results of Spider3 and both the PSSMs of PSI-BLAST and HHBlits.

In the original manuscript, we tested the feasibility of SSE-PSSM using third-party datasets TS115 and CASP12. In the revision, CASP13 and TSsspro8 (please see below) were also utilized. TSsspro8 is composed of PDB structures deposited between May 2012 and Aug. 2013, TS115 and CASP12 datasets comprised structures deposited after Jan. 2016 that were highly non-redundant from structures before 2016, and CASP13 contained novel structures deposited mainly between 2017 and 2018. Because the seven PSI-BLAST-based SSP algorithms (released before 2016) and the four newly-applied HHBlits-based algorithms (released after 2017) should have all been trained with structures deposited before 2013, they all performed well on TSsspro8, no matter in the accuracy pretest (S5 Table) or our feature integration experiments (S2 and S4 Tables). As for TS115, CASP12, and CASP13, the accuracy of the algorithms decreased as the dataset became increasingly updated (i.e., containing more far-related proteins from the training datasets of the algorithms). Perhaps because the HHBlits-based algorithms had been sufficiently trained with proteins deposited before 2016, their accuracy on TS115 and CASP12 was remarkably higher than on CASP13. For instance, the micro-average Q3 of NetSurfP-2, MUFOLD-SS, Porter5, and Spider3 on TS115 and CASP12 was 0.840 and 0.860, respectively, and their micro-average Q3 on CASP13 was only 0.825 (S5 Table).

Regardless of the performance of individual algorithms on any dataset, we had previously shown that integrating the SSE-PSSM can improve the accuracy of all the seven PSI-BLAST-based algorithms (the old Table 5; update to be S2 Table). Thanks to the reviewer's suggestion, this conclusion is further verified using a more recent independent dataset (CASP13) and HHBlits-based algorithms (Figs 2 and 3). In the revised manuscript, we have marked the contents about the new datasets with orange and new algorithms with light green text background colors, respectively.

5) This method essentially is a template-based method (remote-homology). Please explicitly say so.

## Response:

We appreciate this advice and have accordingly done it. In the first paragraph of the revised Discussion (Page 19), we stated, "Since the critical step of the SSE-PSSM algorithm is to determine the SSE sequences of the query protein's homologs based on a reference structure dataset, an SSP method developed using the SSE-PSSM is essentially a template-based method."

Reviewer #2:

The authors have presented a secondary structure prediction algorithm using a machine learning model trained on PDB structures with DSSP SSE assignments.

The mapping of secondary structure information on pairwise sequence alignment to generate a position specific scoring matrix is very simple to understand and is presented in an intelligible fashion.

Although, the use of local pairwise alignment for structure template based algorithms is very risky and can prove to be fatal yet almost every other template based sequence to structure predictor does exactly that. Therefore, I shall not comment over such fundamental ideological discrepancies amongst sequence and structural specialists.

## Response:

We would like to thank the reviewer for this pertinent comment. We had also realized that using pairwise alignment to determine the template for SSP might be inadequate, especially when the query is novel. This is why the proposed SSP feature set was a position specific scoring matrix (PSSM), the establishment of which relied on the alignments between the query sequence and many homologs. Nevertheless, the way we implemented the SSE-PSSM was still based on pairwise alignments. As we verified the compatibility of the SSE-PSSM with recent SSP algorithms such as MUFOLD-SS and NetSurfP-2, which all relied on the PSSM generated by HHBlits, we got the chance to implement the SSE-PSSM based on multiple alignments because the HHBlits readily provides the multiple sequence alignment (MSA) between the query protein and identified homologs. Comparing the SSE-PSSMs generated by PSI-BLAST pairwise alignments and HHBlits MSA, the latter achieved higher Q3/8 accuracies (compare the SSE-PSSM in S1 Table and the SSE-PSSMhh in S4 Table), especially when it was integrated with the amino acid PSSM. Thanks to the comment of Reviewer 2 and the question raised by Reviewer 1, we realized the power of MSA for enhancing the proposed SSP feature set. In the revised manuscript, descriptions of the MSA-based SSE-PSSM are also marked with a light green text background.

Overall, the article is well explained and does appear to be lengthy to read and comprehend. This is a major drawback in the manuscript organization. I would like to suggest the authors to cut-short the length of the article for convenient reading. Longer articles tend to have lesser citations.

## Response:

We would like to thank the reviewer for the positive comments and helpful suggestion. Please see the last part of this letter for modifications made to cut short the manuscript.

Before this article can be accepted to be published, I have some major concerns in the designed study and reported statistics. Concerns are listed as follows:

## Response:

Sincerely, we thank the reviewer for carefully reviewing our work. The questions and information related to the listed concerns provided valuable materials for us to test the proposed SSP feature set. The questions are answered point-to-point as follows.

Major_1: In Table 6 the authors have tried to show a benchmark (control) analysis by running the known SoA SSP methods over standardized data-set after tinkering their protocol with PSI-BLAST v2.3.0.

1) Authors had used a version of 2015 and not a recent one. While I would like to know the rationale behind the choice, my important concern is that:

The references listed as no. 6 and 7 were published in 2014 and 2016 resp. and used the same or perhaps an older version of PSI-BLAST.

## Ans:

We have long been using PSI-BLAST in research and teaching. As this project began in 2016, the PSI-BLAST we had was v2.3.0 (release date: Nov. 30, 2015). The SSP methods applied in this work might be developed with different versions of PSI-BLAST, but most authors did not tell the version they used (inclusive of the two references mentioned by the reviewer). Therefore, we used the one we already had at hand. Besides, to make fair assessments and prevent unexpected influences, we thought that the version and parameter settings of PSI-BLAST should be regarded as "control variables" of the experiments and remain constant. This is why we fixed the PSI-BLAST version/parameters of all SSP programs (S8 Table) and made the performance pretest (updated as Table 4).

The NCBI-BLAST package updates from v2.2.28 to v2.3.0 does not list any major change to PSI-BLAST pipeline. Did the improvements made to HSP calculation and/or handling of compositional biases in BLAST, impacted the difference between the performances of SSpro8, DeepCNF as listed in the references and as reported in this study?

## Ans:

It would be challenging to examine which part of the PSI-BLAST algorithm influenced the performance of the SSP methods. Since it was only utilized as a PSSM generator in the pipeline of conventional SSP programs, most authors did not provide the parameter settings they applied. However, we speculated that the difference in performance shown in Table 4 (the old Table 6) basically resulted from parameter settings. In one of our recent SSP studies, we found that the E-value cutoff dramatically affects the accuracy of SSpro8. The default cutoff of SSpro8 was 0.001. Changing the cutoff to 10 can significantly improve the accuracy (please see the Fig 1 of [73]). The E-value cutoff of PSI-BLAST applied in this study was 100. Second, we had also computed the accuracy of DeepCNF on TS115 and CASP12 using its default PSI-BLAST settings; the Q3 was 0.822 and 0.820, respectively, merely 0.001 lower than the Q3 reported in [14]. Another factor that might cause the observed difference in Table 4 is the target dataset, as we had also found that the size and composition of the target dataset can influence accuracy [73]. Following the regular evaluation procedure of PSI-BLAST-based SSP methods and the clues provided by [14], we used the UniRef90 released in 2015 as the target dataset. However, the actual dataset applied to assess the SSP methods was not stated in [14]. The UniRef90-2015 we used was released in December, but there were 11 other releases in 2015. If the same target dataset could be used, we should be able to obtain the same accuracy reported in [14].

Because of the reviewer's question, we felt interested in to what extent the version of PSI-BLAST would influence the accuracy of SSP methods. We have tested several versions and found that the accuracies registered by ≥v2.2.28 were nearly the same.

----------

TS115/CASP12, v.2.2.21, v.2.2.28, v2.3.0, v2.6.0, v2.11.0

----------

DeepCNF, Q3, 0.825/0.815, 0.825/0.817, 0.826/0.819, 0.825/0.817, 0.825/0.817

SSpro8, Q3, 0.701/0.680, 0.797/0.773, 0.798/0.771, 0.797/0.773, 0.797/0.773

DeepCNF, Q8, 0.717/0.715, 0.716/0.714, 0.716/0.714, 0.716/0.714, 0.716/0.714

SSpro8, Q8, 0.571/0.560, 0.686/0.664, 0.686/0.664, 0.686/0.664, 0.686/0.664

----------

When an older version was applied, the accuracy of SSpro8 significantly decreased. The above is a list of the macro-average Q accuracies of DeepCNF and SSpro8 on TS115/CASP12. The full results of our test on PSI-BLAST versions can be found in S5 Table. Although it may be difficult to explain the underlying mechanism for these results, our test showed that, unless the version of PSI-BLAST is very old, the performance of the applied SSP programs is stable.

2) The authors have deliberately tried to cite SSPro8's performance from listed ref. 7(DeepCNF's Sci. Reports, 2016) and DeepCNF's accuracy from listed ref. 10 (2018 review) rather than using the values from their own studies [listed ref. 6 and 7]. If the authors used the accuracy values from SSPro8's and DeepCNF references then their accuracy should be 87.92% and 85.40% instead of 79.5% and 82.3% resp.

## Ans:

The PSI-BLAST parameter settings and the choice of target dataset can remarkably influence the performance of an SSP algorithm [73]. Even when the parameters and target dataset remain unchanged for an algorithm, different accuracy will still be produced for different query sets. For instance, the accuracies of DeepCNF on TS115 and CASP12 are different under the same experimental conditions. Thus, comparing SSP methods using accuracy data obtained from individual papers based on different experimental conditions would be very difficult.

In order to make the accuracy of various SSP algorithms comparable, we tried our best to use just one source of performance data. We cited the accuracy of DeepCNF and four other algorithms from [14] (2018 review) because all those algorithms were assessed with the same target and query datasets (UniRef90-2015 vs. TS115 and CASP12). For comparability as well, we used those target and query datasets to perform all experiments in our study. The exception in Table 4 were the accuracies of SSpro8 and RaptorX, which were obtained from [7] (DeepCNF) but not [14]. The only reason was that they were not computed in [14]. As for the reason why we cited these missing accuracies from [7] instead of the paper of SSpro8 (or RaptorX), we had stated in the main text, "using … TS115 and CASP12 … and compared the results with previous reports on the same methods using equivalent datasets [7,14]." Regarding SSpro8 and RaptorX, the equivalent datasets were the CullPDB and CASP11 query sets utilized in [7] (noted in the legend of Table 4). The CASP11 was considered equivalent to CAPS12 because they are CASP datasets with close release dates. CASP10 was also tested in [7], but we did not use it because its release date was too far from CAPS12. The CullPDB was a 25% non-redundant dataset of PDB, and it was considered equivalent to TS115 because the latter was a 30% non-redundant dataset of PDB.

Thanks to the question raised by the reviewer, we now find it inappropriate to cite the Q8 of SSpro8 from [7]. Although the Q8 values of SSpro8 were computed in [14], we did not use them previously because they appeared to have only two decimal places (0.xx), and the Q8 listed in [7] had three (0.xxx). Now we realize that citing the data from [14] is more reasonable and have accordingly corrected them in the new Table 4. As for RaptorX, we keep using data from [14] because Q3 was not computed in the paper of RaptorX [5]; besides, the Q8 reported in [5] was assessed with datasets less equivalent to TS115/CASP12 than CullPDB/CASP12.

The reported values for SSPro8 are also misleading as the article reports the SSPro8 values for older template free method while SSPro8 published in 2014 showcases drastic improvements in prediction when employing secondary structure data.

Since this article is specifically about a structure based PSSM, therefore, I expect a clarification on the choices that the authors made to report these values.

## Ans:

The SSE-PSSM algorithm relies on protein secondary structure data and thus could be categorized as a template-based method. However, the SSE-PSSM was not constructed based on any specific template(s). As we designed SSE-PSSM, we speculated that it might have some fundamental differences from classic template-based SSP algorithms. Therefore, the comparison between the SSE-PSSM predictive model and the template-based version SSpro8 (abbreviated as SSpro8T) was not carried out. We would like to thank the reviewer for the constructive comment. After comparing the performance of SSE-PSSM with SSpro8T and performing feature-integration tests between them, we clearly saw their differences. For example, as shown in S2 and S5 Tables, the accuracy of SSpro8T significantly decreased as the independent dataset became more and more updated (meaning increasingly far-related from its training data). The accuracy of the SSE-PSSM predictive models, just like DeepCNF and other template-free methods, decreased much less than SSpro8T did. The feature-integrated tests revealed that SSE-PSSM is capable of improving the accuracy of all assessed template-free methods, but integrating it with SSpro8T exhibited negligible improvements or even adverse effects (S2 Table). Feature-integration tests between PSIPRED/DeepCNF and SSpro8T also showed adverse effects. Please see the first subsection of Discussion for details.

3) The listed ref 6 for SSpro8 claims the secondary structure prediction problem to be essentially solved by showcasing a 92% accuracy using their structure template based approach. While their claim can be disputed, it would be worthwhile to check the performance of SSE-PSSM on the exact same data-set as used in ref. 6.

## Response:

We appreciate the reviewer's suggestion about using the dataset of [6] (SSpro8) to evaluate SSE-PSSM. Since the authors had not released their dataset, we established one following the procedure they made the testing dataset (namely, the pdb_post [6]). The authors said that their testing dataset contained proteins deposited in PDB between May 2012 and Aug. 2013, but the release date of their PDB source data was not stated. Hence, we used the closest PDB release that we had: Jan. 2, 2014. The testing dataset of [6] comprised 11,213 polypeptides solved by X-ray crystallography with ≤2.5 angstroms resolution and possessing ≥30 residues without chain breaks. Based on the same criteria, the testing dataset we prepared contained 10,226 polypeptides and was named TSsspro8 (S3 File). The Q3 and Q8 of SSpro8 on this dataset were 94% and 92%, respectively (S5 Table: Panel C-2).

The TSsspro8 has been used to assess SSP methods in almost every experiment. Because this dataset is old, its data were provided in Supplementary Information files. Here we would like to express our profound apologies to the Editor and Reviewers for their long waiting. Because TSsspro8 is large and several applied SSP algorithms are computationally expensive (for instance, DeepCNF and Poter5 both took ~40 min to predict one protein), every experiment of this revision took several weeks. We just summarized the main results about TSsspro8 (and the template-based SSproo8) in previous paragraphs. In the revised manuscript, statements and Supplementary Information files related to TSsspro8 are marked orange.

Major_2: Authors have not disclosed the quality of PDB structures used. Since, the PSSM is based on DSSP secondary structure assignment, it is imperative for the readers to know:

1. What was the resolution cut-off?

2. Were there any missing regions in the final PDB data-set used, if so, how was it handled?

## Response:

We thank the reviewer for pointing out the information that we should provide. PDB polypeptides with chain breaks or missing regions of any length were eliminated as the nrPDB datasets (S1 and S2 Files) were prepared. Besides, polypeptides shorter than 20 residues were discarded. Because we wanted to test SSP algorithms with real-life cases, we had not restricted the structure-determining method or the resolution of structures. Since all the developmental datasets (training/testing query and target datasets) were sampled from the nrPDB25-2015 (S1 File), we analyzed the resolution propensities of nrPDB25-2015 proteins, as listed below.

----------

Resolution | No. of | Percentage

....(Å)..... | polypeptides.. | in nrPDB25-2015 (%)

----------

N/A (NMR), 707, 6.2

< 1.0, 25, 0.2

1.0 – 2.0, 4062, 35.5

2.0 – 3.0, 5591, 48.9

3.0 – 4.0, 978, 8.5

4.0 – 5.0, 49, 0.4

≥ 5.0, 31, 0.3

----------

Major_3: It is quite intriguing to understand that what motivated the authors to use three and not one sequence homology based clustering approaches to reduce the data-set size. Currently, CD-HIT, USEARCH, and MMseq2 were used to generate nrPDB25-2015 data-set.

## Response:

We feel thankful for the reviewer's asking. The ultimate reason was to ensure the non-redundancy of the nrPDB25-2015, the source of all developmental datasets we used. PLOS ONE is not the first journal we submit this work. In the first version, we used the nrPDB30 prepared by the PDB, and the editor recommended us to make extra homology reduction because the official nrPDBs were prepared by heuristic clustering. After we redid all experiments with USEARCH nr25 reduction and submitted the work to another journal, a reviewer recommended CD-HIT while the other recommended using MMseqs2. Even though redoing all experiments in each revision took us several months, we were actually lucky to receive those suggestions. Perhaps because of the increased diversity of the training data, each time we performed more stringent homology reduction, the accuracy of SSE-PSSM predictive models increased. These experiences had inspired us to seek a general strategy for improving the speed of SSP and led us to discover how the composition of target dataset influences the accuracy of SSP methods. The speed-improving and accuracy-influencing dataset manipulation strategies have recently been published in PLOS ONE [73].

Minor_1:

Reduce the manuscript size by making good use of the supplementary space.

1. Transform data Tables to data-point plots with important data labels visible. The crude tables can be provided in the supplementary files for the curious minded.

## Response:

We thank the reviewer for the helpful suggestions. We have moved the original Tables 4, 5, and 7 as well as Fig 3 into the supplementary space. The original Table 5 was also transformed to a bar chart (the new Fig 2) to make conspicuous the important data. The new data of HHBlits-based SSP methods were also diagrammed in the new Fig 3, and the crude table was provided in the supplementary space.

2. Lots of redundant sentences can be found in the main text. Please remove these.

Ex:

Ln.560-563 and 564- "The experimental protein structural data were obtained from the 30% identity non-redundant dataset released by the Protein Data Bank [72] in 2015 and were then homology-reduced to 25% sequence identity." "All proteins in this source dataset were deposited in the PDB no later than Dec. 2015."

Ln 380-384 is completely irrelevant and not required.

## Response:

We appreciate the reviewer very much for helping us improve the quality of this article. Lines 380–384 of the previous manuscript were removed, so was the Conclusion. We have re-examined the whole article to erase redundancies and simplify the main text, hoping to improve the reader's reading experience. The tracking of Microsoft Word was retained in the "Revised Manuscript with Track Changes" file, showing how extensive the article was edited to cut-short the length.

Suggestion:

After reviewing the article, I strongly encourage the authors to rather develop this methodology as a web-service and compile it for a specialized journal where it shall have its due attention and criticism.

## Response:

Sincere thanks to the reviewer for the encouragement. To facilitate the application of the proposed feature set in SSP and other fields, we have established a freely available web server and standalone programs for generating SSE-PSSM. The web server is still simple but will be continuously refined. When the SSE-PSSM feature set is formally integrated with an advanced SSP algorithm, we will publish a paper on the SSE-PSSM-powered secondary structure predictor.

6. Do you want your identity to be public for this peer review?

Reviewer #1: No

Reviewer #2: No

## Response:

We would like to express again our sincere thanks to the anonymous reviewers for helping us improve this work.

---

## [Decision Letter · Decision Letter 1]

12 Jul 2021

A secondary structure-based position-specific scoring matrix applied to the improvement in protein secondary structure prediction

PONE-D-20-15212R1

Dear Dr. Lo,

We’re pleased to inform you that your manuscript has been judged scientifically suitable for publication and will be formally accepted for publication once it meets all outstanding technical requirements.

Kind regards,

Alexandre G. de Brevern, Ph.D.

Academic Editor

PLOS ONE

Additional Editor Comments (optional):

Reviewers' comments:

Reviewer's Responses to Questions

**Comments to the Author**

1. If the authors have adequately addressed your comments raised in a previous round of review and you feel that this manuscript is now acceptable for publication, you may indicate that here to bypass the “Comments to the Author” section, enter your conflict of interest statement in the “Confidential to Editor” section, and submit your "Accept" recommendation.

Reviewer #2: All comments have been addressed

2. Is the manuscript technically sound, and do the data support the conclusions?

Reviewer #2: Yes

3. Has the statistical analysis been performed appropriately and rigorously? 

Reviewer #2: No

4. Have the authors made all data underlying the findings in their manuscript fully available?

Reviewer #2: Yes

5. Is the manuscript presented in an intelligible fashion and written in standard English?

Reviewer #2: Yes

6. Review Comments to the Author

Reviewer #2: I thank the authors for their painstaking efforts in correcting the manuscript and addressing all of my concerns. I am happy to accept the ms in its current form. However, I would have liked to see the webserver as a part of this article rather than another separate article. It would enhance the impact of the current article and also will provide readers a single point reference for the methodology in the ms.

7. PLOS authors have the option to publish the peer review history of their article (what does this mean?). If published, this will include your full peer review and any attached files.

Reviewer #2: **Yes: **Tarun Jairaj Narwani

---

## [Editor Report · Acceptance letter]

19 Jul 2021

PONE-D-20-15212R1 

A secondary structure-based position-specific scoring matrix applied to the improvement in protein secondary structure prediction 

Dear Dr. Lo:

I'm pleased to inform you that your manuscript has been deemed suitable for publication in PLOS ONE. Congratulations! Your manuscript is now with our production department. 

Kind regards, 

on behalf of

Dr. Alexandre G. de Brevern 

Academic Editor

PLOS ONE